# IBCL: Zero-shot Model Generation for Task Trade-offs in Continual Learning

## Abstract

Like generic multi-task learning, continual learning has the nature of multi-objective optimization, and therefore faces a trade-off between the performance of different tasks. Researchers have discussed how to train models to address user trade-off preferences between the tasks. However, existing algorithms (1) do not guarantee Pareto-optimality and (2) require a training overhead proportional to the number of user preferences, which may be infinite, thus generating a substantial cost. As a response, we propose Imprecise Bayesian Continual Learning (IBCL). Upon a new task, IBCL (1) updates a knowledge base in the form of a convex hull of model parameter distributions and (2) generates one model per given user preference without any additional training. That is, obtaining the user-preferred models via IBCL is zero-shot. Moreover, IBCL ensures a buffer growth that is sublinear in the number of tasks. Experiments show IBCL improves on baseline methods by at most 45% in average per task accuracy and by 43% in peak per task accuracy, while maintaining a near-zero to positive backward transfer. Moreover, its training overhead, measured by number of batch updates per task, does not scale up with the number of preferences. Therefore, it is a well-performing and efficient algorithm when a large number of preferences are requested.

## 1 Introduction

Multi-task learning aims to obtain a shared model to maximize performance on multiple tasks. Therefore, it has the nature of multi-objective optimization (Kendall et al., 2018; Sener and Koltun, 2018), where trade-offs exist between individual tasks. Lifelong machine learning, also known as continual learning (CL), is a special case of multi-task learning where tasks arrive in sequential order (Chen and Liu, 2016; Parisi et al., 2019; Ruvolo and Eaton, 2013b; Thrun, 1998). Formally, the trade-off in CL between not forgetting previous knowledge vs. acquiring new knowledge is known as *stability-plasticity trade-off* (De Lange et al., 2021). So far, researchers have leveraged user preferences to select what points on a trade-off curve should be aimed for during multi-task and continual learning (Mahapatra and Rajan, 2020; Kim et al., 2023).

Consider the following motivating example of a movie recommendation system. The model is first trained to rate movies in the sci-fi genre. Then, a new genre, e.g., documentaries, is added by the movie company. The model needs to learn how to rate documentaries while not forgetting how to rate sci-fis. Training this model boils down to a continual learning problem.

The company now wants to build a recommendation system that adapts to users' tastes in movies. For example, a user Alice tells the company that she has equal preferences over sci-fis and documentaries. Another user Bob says he absolutely wants to watch documentaries and has no interest in sci-fis at all. Consequently, the company's goal is to train two customized models for Alice and Bob respectively, to predict how likely a sci-fi or a documentary should be recommended to an individual user. Based on preferences, Alice's personal model should balance between the accuracy in rating sci-fis and rating documentaries, while Bob's model can compromise the accuracy in rating sci-fis in order to achieve a high accuracy in rating documentaries. As new genres are added, users should be able to input their preferences over all available genres to obtain customized models.

The example above urges us to train Pareto-optimal models under user trade-off preferences between the tasks. State-of-the-art techniques formalize a user's preference as a probability vector over tasks. For instance, if we have two tasks, vector $(0.5, 0.5)^\top$ means that they are equally preferred, while

$(1,0)^\top$ means that the user is only interested in the first one (Mahapatra and Rajan, 2020; 2021). Then, the user-preferred model is trained with a weighted sum of losses on each task's training data, with the weights being the preferences (Lin et al., 2019; 2020; De Lange et al., 2021; Servia-Rodriguez et al., 2021).

However, we identify two major disadvantages in the state-of-the-art approach. First, there is no guarantee of Pareto-optimality. We do not know whether the obtained model is the Pareto-optimal one associated with the given user preference. Aside from the hard Pareto-optimality guarantee, existing techniques do not provide even soft guarantees. For example, they do not inspect how far – with respect to some metric – the obtained model is from the Pareto-optimal one, or how likely it is to be Pareto-optimal under some probability distribution. Second, the training cost is proportional to the number of user preferences. Specifically, we need to train one model per preference, imposing large time and sample overheads, since the number of user preferences can be infinitely large. The need for lowering the training cost urges us to seek few-shot or zero-shot model generation techniques. Although model-based (Finn et al., 2017; Yoon et al., 2018b; Navon et al., 2020; Von Oswald et al., 2019) and prompt-based (Radford et al., 2021; Wang et al., 2022a;b) methods enable few-shot or zero-shot knowledge transfer in continual learning, they have yet to discuss generating models under a specific preference over tasks performance.

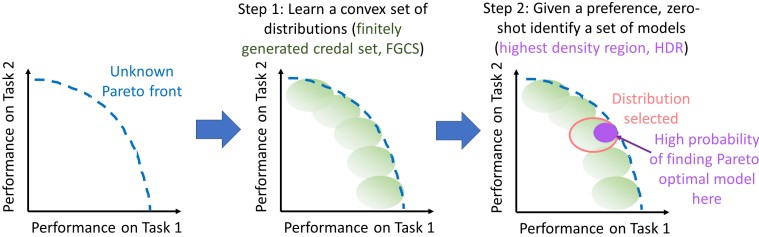

Figure 1: The IBCL workflow, illustrated without loss of generalization on two tasks.

We propose Imprecise Bayesian Continual Learning (IBCL), a Bayesian continual learning algorithm to tackle the two identified shortcomings, with the workflow illustrated in Figure 1. Upon the arrival of a new task's training data, IBCL is trained to update its *knowledge base* (that is, all information shared across tasks), which is a convex set of distributions $\mathcal{Q}$, formally known as a *fintely generated credal set* (FGCS) (Caprio et al., 2023). Each distribution in the FGCS corresponds to one Pareto-optimal model on the trade-off curve, i.e., a model's parameters are viewed as a random variable $\theta$ sampled from a distribution in $\mathcal{Q}$. Since FGCS $\mathcal{Q}$ is convex, we only need to learn and cache its extreme points. Then, given a preference $\bar{w}$, IBCL selects the distribution $\hat{q}_{\bar{w}} \in \mathcal{Q}$ corresponding to the preferred Pareto-optimal model. Once we have $\hat{q}_{\bar{w}}$, we identify a set of model parameters, known as a *highest density region* (HDR), which is the smallest parameter set that contains the Pareto-optimal model with high probability. Detailed definitions of FGCS and HDR can be found in Section 2.

Compared to existing efficient knowledge transfer like Wang et al. (2022b) and Navon et al. (2020), IBCL enables to specify trade-off preferences over tasks. Compared to existing multi-task and continual learning methods that allow to specify preferences (Lin et al., 2020), IBCL does not require training one model per preference. In other words, IBCL significantly improves efficiency by maintaining a constant training overhead per task, regardless of number of preferences requested. Moreover, IBCL probabilistically guarantees to find the Pareto-optimal models under given preferences, a property not achievable by state-of-the-art.

IBCL is a method that belongs to the Bayesian continual learning (BCL) framework (Nguyen et al., 2018) which, by design, mitigates *catastrophic forgetting*, that is, the forgetting part of the stability-plasticity trade-off. This is shown in Appendix A, where we also carry out a more in-depth discussion on BCL, and the relationship between IBCL and other BCL techniques.

**Contributions.** 1. We propose IBCL, a Bayesian continual learning algorithm that (i) probabilistically guarantees to find the Pareto-optimal models associated with user trade-off preferences between the performance over the tasks, (ii) zero-shot generates these models, so that the training overhead per task is constant, regardless of number of preferences, and (iii) has a sublinear buffer growth in the number of tasks. 2. We prove the probabilistic guarantee of Pareto-optimality. 3. We

evaluate IBCL on image classification and NLP benchmarks to support our claims. We find that IBCL improves on baselines by at most 45% in average per task accuracy and by 43% in peak per task accuracy while maintaining a near-zero to positive backward transfer, with a constant training overhead regardless of number of preferences. Ablation studies are also performed.

## 2 BACKGROUND

Our algorithm hinges upon the concepts of finitely generated credal set (FGCS) from Imprecise Probability (IP) theory (Walley, 1991; Troffaes and de Cooman, 2014; Caprio and Gong, 2023; Caprio and Mukherjee, 2023; Caprio and Seidenfeld, 2023; Caprio et al., 2023).

**Definition 1** (Finitely Generated Credal Set). *A convex set $\mathcal{Q} = \{q : q = \sum_{j=1}^{m} \beta^j q^j, \beta_j \geq 0 \, \forall j, \sum_j \beta_j = 1\}$ of probability distributions with finitely many extreme elements $ex[\mathcal{Q}] = \{q^j\}_{j=1}^{m}$ is called a finitely generated credal set (FGCS).*

In other words, an FGCS is a convex hull of (finitely many) distributions. We also borrow from the Bayesian literature the idea of highest density region (HDR) (Coolen, 1992).

**Definition 2** (Highest Density Region). *Let $\Theta$ be a set of interest, and $\alpha \in [0, 1]$ be a significance level. Suppose that a (continuous) random variable $\theta \in \Theta$ has probability density function (pdf) $q$.[1] Then, the $(1 - \alpha)$-HDR is the set $\Theta_q^\alpha$ such that $\int_{\Theta_q^\alpha} q(\theta)d\theta \geq 1 - \alpha$ and $\int_{\Theta_q^\alpha} d\theta$ is minimum.*

Definition 2 tells us that if $\theta \sim q$, $\Pr_{\theta \sim q}[\theta \in \Theta_q^\alpha] \geq 1 - \alpha$, and $\Theta_q^\alpha$ is the narrowest subset of $\Theta$ that guarantees this inequality. In other words, the HDR $\Theta_q^\alpha$ is the smallest subset of $\Theta$ where we can find the realization of random variable $\theta$ with high probability. The concept of HDR is further explained in Appendix B with an illustration. Other related works, including multi-task and continual learning, are reviewed in Appendix C.

## 3 PROBLEM FORMULATION

Our goal is to obtain classification models for domain-incremental learning (Van de Ven and Tolias, 2019) under user preferences over task trade-offs. The models should probabilistically guarantee Pareto-optimality, with training overhead not scaling up with the number of user preferences.

### 3.1 ASSUMPTIONS

Formally, we denote a measurable space by $(\cdot, \mathcal{A}(\cdot))$, i.e. the tuple of a set and a $\sigma$-algebra endowed to it. Let $(\mathcal{X}, \mathcal{A}(\mathcal{X}))$ be the measurable space of data, $(\mathcal{Y}, \mathcal{A}(\mathcal{Y}))$ be the measurable space of labels, and $(\mathcal{X} \times \mathcal{Y}, \mathcal{A}(\mathcal{X} \times \mathcal{Y}))$ be the measurable product space of data and labels. Next, we denote the space of all probability measures on a measurable space by $\Delta(\cdot, \mathcal{A}(\cdot))$. That is, all possible distributions on $\mathcal{X} \times \mathcal{Y}$ belong to set $\Delta(\mathcal{X} \times \mathcal{Y}, \mathcal{A}(\mathcal{X} \times \mathcal{Y}))$, simplified as $\Delta_{\mathcal{X}\mathcal{Y}}$.

A task $i$ is associated with a distribution $p_i \in \Delta_{\mathcal{X}\mathcal{Y}}$, from which labeled data can be i.i.d. drawn.[2] We assume that all tasks are similar to each other.

**Assumption 1** (Task Similarity). *For all task $i$, $p_i \in \mathcal{F}$, where $\mathcal{F}$ is a convex subset of $\Delta_{\mathcal{X}\mathcal{Y}}$. Also, we assume that the diameter of $\mathcal{F}$ is some $r > 0$, that is, $\sup_{F, F' \in \mathcal{F}} \|F - F'\|_{W_2} \leq r$, where $\|\cdot\|_{W_2}$ denotes the 2-Wasserstein distance.*

In an effort to make the paper self-contained, in Appendix D we give the definition of 2-Wasserstein distance, as well as the reason we choose it. Assumption 1 is needed to mitigate the possible model misspecification, which in turn could lead to catastrophic forgetting even when Bayesian inference is carried out exactly. For more, see Kessler et al. (2023) and Appendix E. Under Assumption 1, for any two tasks $i$ and $j$, their underlying distributions $p_i$ and $p_j$ satisfy $\|p_i - p_j\|_{W_2} \leq r$. Moreover, since $\mathcal{F}$ is convex, any convex combination of task distributions belongs to $\mathcal{F}$. Next, we assume the parameterization of class $\mathcal{F}$.

---

[1]Here, for ease of notation, we do not distinguish between a random variable and its realization.
[2]Notice that $p_i$ denotes the pdf/pmf of probability measure $P_i$. In the remainder of the paper, we do not distinguish between them for notational convenience.

**Assumption 2** (Parameterization of Task Distributions). *Every distribution $F$ in $\mathcal{F}$ is parameterized by $\theta$, a parameter belonging to a parameter space $\Theta$.*

An example of a parameterized family that satisfies Assumption 1 is given in Appendix F. Notice that all tasks share the same data space $\mathcal{X}$ and label space $\mathcal{Y}$, so the learning is domain-incremental. We then formalize user preferences over tasks.

**Definition 3** (User Preferences over Tasks). *Consider $k$ tasks with underlying distributions $p_1, p_2, \ldots, p_k$. We express a preference over them via a probability vector $\bar{w} = (w_1, w_2, \ldots, w_k)^\top$, that is, $w_i \geq 0$ for all $i \in \{1, \ldots, k\}$, and $\sum_{i=1}^{k} w_i = 1$.*

Based on this definition, given a user preference $\bar{w}$ over all $k$ tasks encountered, the personalized model for the user aims to learn the distribution $p_{\bar{w}} := \sum_{i=1}^{k} w_i p_i$. It is the distribution associated with tasks $1, \ldots, k$ that also takes into account a preference over them. Since $p_{\bar{w}}$ is the convex combination of $p_1, \ldots, p_k$, thanks to Assumptions 1 and 2, we have that $p_{\bar{w}} \in \mathcal{F}$, and therefore it is also parameterized by some $\theta \in \Theta$.

The learning procedure is the same as standard supervised domain-incremental learning. Upon task $k$, we draw $n_k$ labeled examples i.i.d. from an unknown $p_k$. Then, we are given at least one user preference $\bar{w}$ over the $k$ tasks so far. The data drawn for task $k + 1$ will not be available until we have finished learning models for all user preferences at task $k$.

## 3.2 MAIN PROBLEM

We aim to design a domain-incremental learning algorithm that generates one model per user preference at each task. Given a significance level $\alpha \in [0, 1]$, at a task $k$, the algorithm should satisfy

1. **Probabilistic Pareto-optimality**. Let $\bar{w}$ be a preference over the $k$ tasks. We want to identify the smallest subset of model parameters, $\Theta_{\hat{q}_{\bar{w}}}^\alpha \subset \Theta$ (written as $\Theta_{\bar{w}}^\alpha$ for notational convenience from now on), that the Pareto-optimal parameter $\theta_{\bar{w}}^\star$ (i.e. the ground-truth parameter of $p_{\bar{w}}$) belongs to with high probability, i.e., $\Pr_{\theta_{\bar{w}}^\star \sim \hat{q}_{\bar{w}}}[\theta_{\bar{w}}^\star \in \Theta_{\bar{w}}^\alpha] \geq 1 - \alpha$, under a known $\hat{q}_{\bar{w}}$ over $\Theta$.
2. **Zero-shot preferred model generation**. When there are more than one user preference $\overline{w}_s$, $s \in \{1, \ldots, S\}$, no training is needed for generating model subsets $\Theta_{\bar{w}_s}^\alpha$, for all $s$, i.e. the model generation is zero-shot.
3. **Sublinear buffer growth**. The memory overhead for the entire procedure should be growing sublinearly in the number of tasks.

# 4 IMPRECISE BAYESIAN CONTINUAL LEARNING

## 4.1 FGCS KNOWLEDGE BASE UPDATE

We take a Bayesian continual learning approach, i.e., the parameter $\theta$ of distribution $p_k$ pertaining to task $k$ is viewed as a random variable distributed according to some distribution $q$. At the beginning of the analysis, we specify $m$ many such distributions, $\text{ex}[\mathcal{Q}_0] = \{q_0^1, \ldots, q_0^m\}$. They are the ones that the designer deems plausible – a priori – for parameter $\theta$ of task 1. Upon observing data pertaining to task 1, we learn a set $\mathcal{Q}_1^{tmp}$ of parameter distributions and buffer them as extreme points $\text{ex}[\mathcal{Q}_1]$ of the FGCS $\mathcal{Q}_1$ corresponding to task 1. We proceed similarly for the successive tasks $i \geq 2$.

In Algorithm 1, at task $i$, we learn $m$ posteriors $q_i^1, \ldots q_i^m$ by variational inference from buffered priors $q_{i-1}^1, \ldots q_{i-1}^m$ one-by-one (line 3). However, we do not want to buffer all learned posteriors, so we use a distance threshold to exclude posteriors with similar distributions to the ones that are already buffered (line 4 - 9). When a distribution similar to $q_i^j$ is found in the knowledge base, we remember to use it in place of $q_i^j$ in the future (line 8). The posteriors without a similar distribution buffered are appended to the knowledge base (line 11).

We note in passing that another reason for requiring Assumption 1 is that, without it, the variational approximation in line 3 may incur catastrophic forgetting.

---

**Algorithm 1** FGCS Knowledge Base Update

---

**Input**: Current knowledge base in the form of FGCS extreme points
$\mathrm{ex}[\mathcal{Q}_{i-1}] = \{q_{i-1}^1, \ldots, q_{i-1}^m\}$, observed labeled data $(\bar{x}_i, \bar{y}_i)$ at task $i$, and a distribution distance threshold $d \geq 0$

**Output**: Updated extreme elements $\mathrm{ex}[\mathcal{Q}_i]$

1: $\mathcal{Q}_i^{tmp} \leftarrow \emptyset$
2: **for** $j \in \{1, \ldots, m\}$ **do**
3: $\quad q_i^j \leftarrow \mathsf{variational\_inference}(q_{i-1}^j, \bar{x}_i, \bar{y}_i)$
4: $\quad d_i^j \leftarrow \min_{q \in \mathrm{ex}[\mathcal{Q}_{i-1}]} \|q_i^j - q\|_{W_2}$
5: $\quad$ **if** $d_i^j \geq d$ **then**
6: $\quad\quad \mathcal{Q}_i^{tmp} \leftarrow \mathcal{Q}_i^{tmp} \cup \{q_i^j\}$
7: $\quad$ **else**
8: $\quad\quad$ Remember to use $q = \arg\min_{q \in \mathrm{ex}[\mathcal{Q}_{i-1}]} \|q_i^j - q\|_{W_2}$ in place of $q_i^j$ later on
9: $\quad$ **end if**
10: **end for**
11: $\mathrm{ex}[\mathcal{Q}_i] \leftarrow \mathrm{ex}[\mathcal{Q}_{i-1}] \cup \mathcal{Q}_i^{tmp}$

---

Notice that Algorithm 1 ensures **sublinear buffer growth** in our problem formulation because at each task $i$ we only buffer $m_i$ new posterior models, with $0 \leq m_i \leq m$. With sufficiently large threshold $d$, the buffer growth can become constant after several tasks. The use of different threshold $d$'s is discussed in our ablation studies, see section 5.2.

### 4.2 ZERO-SHOT GENERATION OF USER PREFERRED MODELS

Next, after we update the FGCS extreme points for task $i$, we are given a set of user preferences. For each preference $\bar{w}$, we need to identify the Pareto-optimal parameter $\theta_{\bar{w}}^\star$ for the preferred data distribution $p_{\bar{w}}$. This procedure can be divided into two steps as follows.

First, we find the parameter distribution $\hat{q}_{\bar{w}}$ via a convex combination of the extreme points in the knowledge base, whose weights correspond to the entries of preference vector $\bar{w}$. That is,

$$\hat{q}_{\bar{w}} = \sum_{k=1}^{i} \sum_{j=1}^{m_k} \beta_k^j q_k^j, \quad \text{where } \sum_{j=1}^{m_k} \beta_k^j = w_k, \text{ and } \beta_k^j \geq 0, \text{ for all } j \text{ and all } k. \quad (1)$$

Here, $q_k^j$ is a buffered extreme point of FGCS $\mathcal{Q}_k$, i.e. the $j$-th parameter posterior of task $k$. The weight $\beta_k^j$ of this extreme point is decided by preference vector entry $\bar{w}_j$. In implementation, if we have $m_k$ extreme points stored for task $k$, we can choose equal weights $\beta_k^1 = \cdots = \beta_k^m = w_k/m_k$. For example, if we have preference $\bar{w} = (0.8, 0.2)^\top$ on two tasks so far, and we have two extreme points per task stored in the knowledge base, we can use $\beta_1^1 = \beta_1^2 = 0.8/2 = 0.4$ and $\beta_2^1 = \beta_2^2 = 0.2/2 = 0.1$.

As we can see from the following theorem, distribution $\hat{q}_{\bar{w}}$ is a parameter posterior corresponding to a preference elicitation via preference vector $\bar{w}$ over the tasks encountered so far.

**Theorem 4** (Selection Equivalence). *Selecting a precise distribution $\hat{q}_{\bar{w}}$ from $\mathcal{Q}_i$ is equivalent to specifying a preference weight vector $\bar{w}$ on $p_1, \ldots, p_i$.*

Please refer to Appendix G for the proof. Theorem 4 entails that the selection of $\hat{q}_{\bar{w}}$ in Algorithm 2 is related to the correct parameterization of $p_{\bar{w}} \in \Delta_{\mathcal{X}\mathcal{Y}}$.

Second, we compute the HDR $\Theta_{\bar{w}}^\alpha \subset \Theta$ from $\hat{q}_{\bar{w}}$. This is implemented via a standard procedure that locates a region in the parameter space whose enclosed probability mass is (at least) $1 - \alpha$, according to $\hat{q}_{\bar{w}}$. This procedure can be routinely implemented, e.g., in R, using package HDInterval (Juat et al., 2022). As a result, we locate the smallest set of parameters $\Theta_{\bar{w}}^\alpha \subset \Theta$ associated with the preference $\bar{w}$. This subroutine is formalized in Algorithm 2, and one remark is that it does not require any training, i.e., we meet our goal of **zero-shot preferred model generation** of section 3.2.

---

**Algorithm 2** Preference HDR Computation

**Input**: Knowledge base $\mathrm{ex}[\mathcal{Q}_i]$ with $m_k$ extreme points saved for task $k \in \{1, \ldots, i\}$, preference vector $\bar{w}$ on the $i$ tasks, significance level $\alpha \in [0, 1]$
**Output**: HDR $\Theta^\alpha_{\bar{w}} \subset \Theta$

1: **for** $k = 1, \ldots, i$ **do**
2:     $\beta^1_k = \cdots = \beta^m_k \leftarrow w_k / m_k$
3: **end for**
4: $\hat{q}_{\bar{w}} = \sum^i_{k=1} \sum^{m_k}_{j=1} \beta^j_k q^j_k$
5: $\Theta^\alpha_{\bar{w}} \leftarrow \mathsf{hdr}(\hat{q}_{\bar{w}}, \alpha)$

---

## 4.3 Overall IBCL Algorithm and Analysis

From the two subroutines in Sections 4.1 and 4.2, we construct the overall IBCL algorithm as in Algorithm 3.

---

**Algorithm 3** Imprecise Bayesian Continual Learning

**Input**: Prior distributions $\mathrm{ex}[\mathcal{Q}_0] = \{q^1_0, \ldots, q^m_0\}$, hyperparameters $\alpha$ and $d$
**Output**: HDR $\Theta^\alpha_{\bar{w}}$ for each given preference $\bar{w}$ at each task $i$

1: **for** task $i = 1, 2, \ldots$ **do**
2:     $\bar{x}_i, \bar{y}_i \leftarrow$ sample $n_i$ labeled data points i.i.d. from $p_i$
3:     $\mathrm{ex}[\mathcal{Q}_i] \leftarrow \mathsf{fgcs\_knowledge\_base\_update}(\mathrm{ex}[\mathcal{Q}_{i-1}], \bar{x}_i, \bar{y}_i, d)$        ▷ % Algorithm 1 %
4:     **while** user has a new preference **do**
5:        $\bar{w} \leftarrow$ user input
6:        $\Theta^\alpha_{\bar{w}} \leftarrow \mathsf{preference\_hdr\_computation}(\mathrm{ex}[\mathcal{Q}_i], \bar{w}, \alpha)$        ▷ % Algorithm 2 %
7:     **end while**
8: **end for**

---

For each task, in line 3, we use Algorithm 1 to update the knowledge base by learning $m$ posteriors from the current priors. Some of these posteriors will be cached and some will be substituted by a previous distribution in the knowledge base. In lines 4-6, upon a user-given preference over all tasks so far, we obtain the HDR of the model associated with preference $\bar{w}$ with zero-shot via Algorithm 2. Notice that this HDR computation does not require the initial priors $\mathrm{ex}[\mathcal{Q}_0]$, so we can discard them once the posteriors are learned in the first task. The following theorem ensures that IBCL locates the user-preferred Pareto-optimal model with high probability.

**Theorem 5** (Probabilistic Pareto-optimality). *Pick any* $\alpha \in [0, 1]$. *The Pareto-optimal parameter* $\theta^\star_{\bar{w}}$, *i.e., the ground-truth parameter for* $p_{\bar{w}}$, *belongs to* $\Theta^\alpha_{\bar{w}}$ *with probability at least* $1 - \alpha$ *under distribution* $\hat{q}_{\bar{w}}$. *In formulas,* $\mathrm{Pr}_{\theta^\star_{\bar{w}} \sim \hat{q}_{\bar{w}}}[\theta^\star_{\bar{w}} \in \Theta^\alpha_{\bar{w}}] \geq 1 - \alpha$.

Theorem 5 gives us a $(1 - \alpha)$-guarantee in obtaining Pareto-optimal models for given task trade-off preferences. Consequently, the IBCL algorithm enjoys the **probabilistic Pareto-optimality** targeted by our main problem. Please refer to Appendix G for the proof.

## 5 Experiments

### 5.1 Setup

We compare IBCL to the following continual learning baselines.

1. **Rehearsal-based.** Rehearsal-based methods memorize a subset of training data at each task. The loss function is a sum of losses on each task's data retained. A preference can be specified as weights when computing the sum (Lin et al., 2019). We choose GEM (Lopez-Paz and Ranzato, 2017) and A-GEM (Chaudhry et al., 2018).

2. **Rehearsal-based, Bayesian.** Rehearsal-based methods are in general deterministic. Since IBCL is Bayesian, we also compare to Bayesian methods and we choose VCL (Nguyen et al., 2018). We equip VCL with episodic memory to make it rehearsal-based to specify a preference. This approach has been used in Servia-Rodriguez et al. (2021).

3. **Prompt-based.** Prompt-based continual learning are considered efficient because they fine-tune a data prefix (prompt) instead of a model at each task. So far, there has been no discussion on how to specify task preferences in these methods. We choose L2P (Wang et al., 2022b), with an attempt to specify preferences by training one prompt per task, and use a preference-weighted sum of the prompts at inference time.

We experiment on four standard continual learning benchmarks, including three image classification and one NLP: (i) 15 tasks in CelebA (Liu et al., 2015) (with vs. without attributes), (ii) 10 tasks in Split CIFAR-100 (Zenke et al., 2017) (animals vs. non-animals), (iii) 10 tasks in TinyImageNet (Le and Yang, 2015) (animals vs. non-animals) and (iv) 5 tasks in 20NewsGroup (Lang, 1995) (news related to computers vs. not related to computers). For the first three image benchmarks, features are first extracted by ResNet-18 (He et al., 2016), and for 20NewsGroup, features are extracted by TF-IDF (Aizawa, 2003).

As in standard continual learning evaluation, after training on task $i$, we evaluate the accuracy on all testing data of previous tasks $j \in \{1, \dots, i\}$. To evaluate how well does a model address preferences, we randomly generate $n_{\text{prefs}} = 10$ preferences per task, except for task 1, whose preference is always given by scalar 1. Therefore, for each method, we obtain 10 models at each task, and we evaluate a preference-weighted sum of their accuracies on previous tasks. Finally, these preference-weighted accuracies are used to compute standard continual learning metrics: average per task accuracy, peak per task accuracy, and backward transfer (Díaz-Rodríguez et al., 2018). Experiments are run on Intel(R) Core(TM) i7-8550U CPU @ 1.80GHz. Detailed setup can be found in Appendix H.1.

## 5.2 RESULTS

Our results support the claim that IBCL not only achieves high performance by probabilistic Pareto-optimality, but is also efficient with zero-shot generation of models.

Since VCL and IBCL output probabilistic models (BNNs and HDRs), we sample 10 deterministic models from each and compute the range of their performance metrics, illustrated as shaded areas in Figures 2 and 3. They represent performances on the 20NewsGroup and TinyImageNet, respectively. In these figures, we draw the curves of top performance and mean performance of the sampled deterministic models by VCL and IBCL as solid and dashed lines, respectively. Due to page limit, we show the results on Split CIFAR-100 and CelebA in Figure 8 and 9 in Appendix H.2.

From Figures 2, 3, 8 and 9, we can see that IBCL overall generates the model with top performance (high accuracy) in all cases, while maintaining little catastrophic forgetting (near-zero to positive backward transfer). This is due to the probabilistic Pareto-optimality guarantee. Statistically, IBCL improves on the baselines by at most 45% in average per task accuracy and by 43% in peak per task accuracy (compared to L2P in 20News). So far, to our knowledge, there is no discussion on how to specify a task trade-off preference in prompt-based continual learning, and we make an attempt by using a preference-weighted sum of all learned prompts in L2P. We can see how this approach generally works poorly, except for CelebA, where L2P performs nearly well as IBCL. We believe the performance by prompts trained in L2P depends on its frozen model, and how to use prompt-based methods to generate preference-specified models is still an open problem.

As illustrated in the figures, IBCL has a slightly negative backward transfer in the very beginning but then this value stays near-zero or positive. This shows that although IBCL may slightly forget the knowledge learned from the first task at the second task, it steadily retains knowledge afterwards. This may be due to the choice of the priors, of the likelihood, of the variational method to approximate the posterior, or to an intrinsic characteristic of our method. Given its relevance, we defer studying this phenomenon to future work. We can also see how, although VCL's backward transfer is higher than IBCL's in the first few tasks, it eventually decreases and takes values that are nearly identical to IBCL ones. For 20NewsGroup, this happens after 5 tasks, for the other datasets after 10.

Table 1 shows the training overhead comparison measured in number of batch updates per task. We can see how IBCL's overhead is independent of the number of preferences $n_{\text{prefs}}$ because it only requires training for the FGCS but not for the preferred models. Therefore, IBCL maintains a constant training overhead regardless of number of preferences. Although prompt-based methods like L2P can also achieve this efficiency, IBCL has a larger overhead only by a constant number of

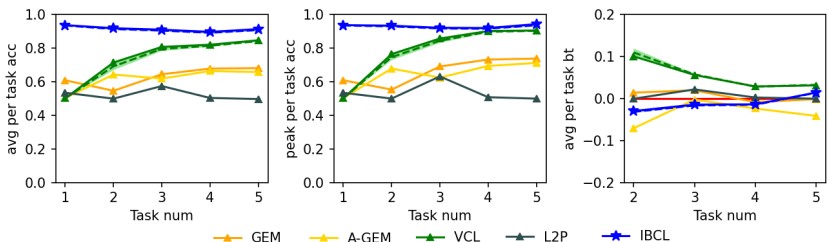

Figure 2: Results of 20NewsGroup. Since VCL and IBCL produce probabilistic models, we sample 10 deterministic models for each. The solid blue curve illustrates the top performance of deterministic models by IBCL, and the dashed blue curve is the mean performance. The shaded blue region is the performance range by IBCL. The same illustration method is used for VCL in green color.

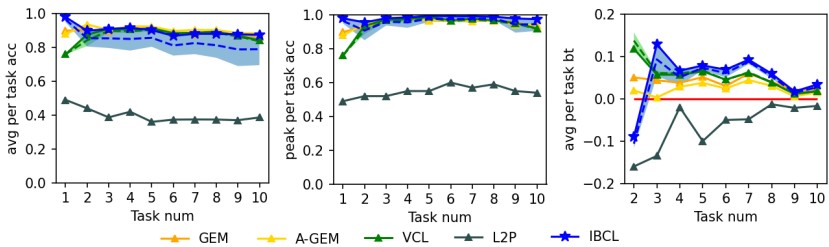

Figure 3: Results of TinyImageNet. The illustration method is the same as in Figure 2.

priors, not to mention that L2P has not yet developed a way to learn preference-specified models and therefore performs poorly.

The main experiments are conducted with hyperparameters $\alpha = 0.01$ and $d = 0.002$. We also conduct two ablation studies. The first one is on different significance level $\alpha$ in Algorithm 2.

In Figure 4, we evaluate testing accuracy on three different $\alpha$'s over five different preferences (from $[0.1, 0.9]$ to $[0.9, 0.1]$) on the first two tasks of 20NewsGroup. For each preference, we uniformly sample 200 deterministic models from the HDR. We use the sampled model with the maximum L2 sum of the two accuracies to estimate the Pareto optimality under a preference. We can see that, as $\alpha$ approaches 0, we tend to sample closer to the Pareto front. This is because, with a smaller $\alpha$, HDRs becomes wider and we have a higher probability to sample Pareto-optimal models according to Theorem 5. For instance, when $\alpha = 0.01$, we have a probability of at least 0.99 that the Pareto-optimal solution is contained in the HDR.

We then evaluate the three $\alpha$'s in the same way as in the main experiments, with 10 randomly generated preferences per task. Figure 5 shows that the performance drops as $\alpha$ increases, because we are more likely to sample poorly performing models from the HDR.

The second ablation study is on different thresholds $d$ in Algorithm 1. As $d$ increases, we are allowing more posteriors in the knowledge base to be reused. This will lead to memory efficiency

| | | # batch updates at task $i$ | # batch updates at last task | | | |
|---|---|---|---|---|---|---|
| | | | CelebA | CIFAR100 | TImgNet | 20News |
| Rehearsal | GEM | $n_{\text{prefs}}$ | 99747 | 19532 | 13594 | 35313 |
| | A-GEM | $\times (n_i + (i-1) \times n_{\text{mem}})$ | | | | |
| | VCL | $\times e/b$ | | | | |
| Prompt | L2P | $n_i \times e/b$ | 9538 | 1250 | 938 | 2907 |
| IBCL (ours) | | $n_{\text{priors}} \times n_i \times e/b$ | 28614 | 3750 | 2814 | 8721 |

Table 1: Training overhead comparison, measured as # of batch updates required at a task. Here, $n_i$: # of training data points at task $i$, $n_{\text{prefs}}$: # of preferences per task, $n_{\text{mem}}$: # of data points memorized per task in rehearsal, $n_{\text{priors}}$: # of priors in IBCL, $e$: # of epochs and $b$: batch size.

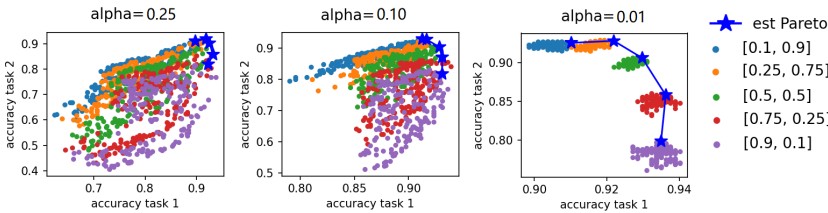

Figure 4: Different $\alpha$'s on different preferences over the first two tasks in 20NewsGroup.

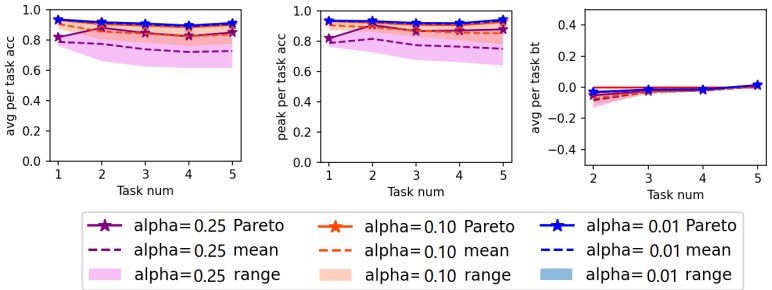

Figure 5: Different $\alpha$'s on randomly generated preferences over all tasks in 20NewsGroup.

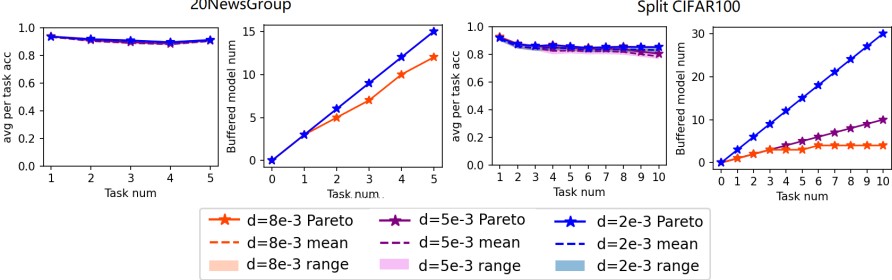

Figure 6: Different $d$'s on 20NewsGroup and Split-CIFAR100. The buffer growth curves of $d = 5e - 3$ and $d = 2e - 3$ of 20NewsGroup are overlapping.

at the cost of a performance drop. Figure 6 supports this trend. We can see how performance barely drops by reusing posteriors, while the buffer growth speed becomes sublinear. For Split-CIFAR100, when $d = 8e - 3$, the buffer size stops growing after task 6.

## 6 DISCUSSION AND CONCLUSION

**Advantages of IBCL.** IBCL (i) guarantees that the Pareto-optimal model under a given preference can be sampled from the output HDR with high probability and (ii) zero-shot generates these preferred model, with training overhead not scaling up with number of preferences. From the experiments, we can see how baseline methods with performance close to IBCL (rehearsal-based) are inefficient, and efficient methods (prompt-based) do not perform well in finding Pareto optimality.

**Limitations of IBCL.** Poorly performing models can also be sampled from IBCL's HDRs. However, in practice, we can fine-tune $\alpha$ to shrink down the HDR to avoid poorly performing ones, as shown in the ablation studies.

Overall, we propose a probabilistic continual learning algorithm, namely IBCL, to locate models for particular task trade-off preferences with probabilistic Pareto-optimality via zero-shot. This means that the training overhead does not scale up with the number of preferences, significantly reducing the computational cost when there is a large number of preferences.

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

## A  REASON TO ADOPT A BAYESIAN CONTINUAL LEARNING APPROACH

Let $q_0(\theta)$ be our prior pdf/pmf on parameter $\theta \in \Theta$ at time $t = 0$. At time $t = 1$, we collect data $(\bar{x}_1, \bar{y}_1)$ pertaining to task 1, we elicit likelihood pdf/pmf $\ell_1(\bar{x}_1, \bar{y}_1 \mid \theta)$, and we compute $q_1(\theta \mid \bar{x}_1, \bar{y}_1) \propto q_0(\theta) \times \ell_1(\bar{x}_1, \bar{y}_1 \mid \theta)$. At time $t = 2$, we collect data $(\bar{x}_2, \bar{y}_2)$ pertaining to task 2 and we elicit likelihood pdf/pmf $\ell_2(\bar{x}_2, \bar{y}_2 \mid \theta)$. Now we have two options.

(i) Bayesian Continual Learning (BCL): we let the prior pdf/pmf at time $t = 2$ be the posterior pdf/pmf at time $t = 1$. That is, our prior pdf/pmf is $q_1(\theta \mid \bar{x}_1, \bar{y}_1)$, and we compute $q_2(\theta \mid \bar{x}_1, \bar{y}_1, \bar{x}_2, \bar{y}_2) \propto q_1(\theta \mid \bar{x}_1, \bar{y}_1) \times \ell_2(\bar{x}_2, \bar{y}_2 \mid \theta) \propto q_0(\theta) \times \ell_1(\bar{x}_1, \bar{y}_1 \mid \theta) \times \ell_2(\bar{x}_2, \bar{y}_2 \mid \theta)$;[3]

(ii) Bayesian Isolated Learning (BIL): we let the prior pdf/pmf at time $t = 2$ be a generic prior pdf/pmf $q_0'(\theta)$. We compute $q_2'(\theta \mid \bar{x}_2, \bar{y}_2) \propto q_0'(\theta) \times \ell_2(\bar{x}_2, \bar{y}_2 \mid \theta)$. We can even re-use the original prior, so that $q_0' = q_0$.

As we can see, in option (i) we assume that the data generating process at time $t = 2$ takes into account both tasks, while in option (ii) we posit that it only takes into account task 2. Denote by $\sigma(X)$ the sigma-algebra generated by a generic random variable $X$. Let also $Q_2$ be the probability measure whose pdf/pmf is $q_2$, and $Q_2'$ be the probability measure whose pdf/pmf is $q_2'$. Then, we have the following.

**Proposition 1.** Posterior probability measure $Q_2$ can be written as a $\sigma(\bar{X}_1, \bar{Y}_1, \bar{X}_2, \bar{Y}_2)$-measurable random variable taking values in $[0, 1]$, while posterior probability measure $Q_2'$ can be written as a $\sigma(\bar{X}_2, \bar{Y}_2)$-measurable random variable taking values in $[0, 1]$.

*Proof.* Pick any $A \subset \Theta$. Then, $Q_2[A \mid \sigma(\bar{X}_1, \bar{Y}_1, \bar{X}_2, \bar{Y}_2)] = \mathbb{E}_{Q_2}[\mathbb{1}_A \mid \sigma(\bar{X}_1, \bar{Y}_1, \bar{X}_2, \bar{Y}_2)]$, a $\sigma(\bar{X}_1, \bar{Y}_1, \bar{X}_2, \bar{Y}_2)$-measurable random variable taking values in $[0, 1]$. Notice that $\mathbb{1}_A$ denotes the indicator function for set $A$. Similarly, $Q_2'[A \mid \sigma(\bar{X}_2, \bar{Y}_2)] = \mathbb{E}_{Q_2'}[\mathbb{1}_A \mid \sigma(\bar{X}_2, \bar{Y}_2)]$, a $\sigma(\bar{X}_2, \bar{Y}_2)$-measurable random variable taking values in $[0, 1]$. This is a well-known result in measure theory (Billingsley, 1986). $\square$

Of course Proposition 1 holds for all $t \geq 2$. Recall that the sigma-algebra $\sigma(X)$ generated by a generic random variable $X$ captures the idea of information encoded in observing $X$. An immediate corollary is the following.

**Corollary 2.** Let $t \geq 2$. Then, if we opt for BIL, we lose all the information encoded in $\{(\bar{X}_i, \bar{Y}_i)\}_{i=1}^{t-1}$.

---

[3]Here we tacitly assume that the likelihoods are independent.

In turn, if we opt for BIL, we obtain a posterior that is not measurable with respect to $\sigma(\{(\bar{X}_i, \bar{Y}_i)\}_{i=1}^t) \setminus \sigma(\bar{X}_t, \bar{Y}_t)$. If the true data generating process $p_t$ is a function of the previous data generating processes $p_{t'}$, $t' \leq t$, this leaves us with a worse approximation of the "true" posterior $Q^{\text{true}} \propto Q_0 \times p_t$.

The phenomenon in Corollary 2 is commonly referred to as *catastrophic forgetting*. Continual learning literature is unanimous in labeling catastrophic forgetting as undesirable – see e.g. Farquhar and Gal (2019); Li et al. (2020). For this reason, in this work we adopt a BCL approach. In practice, we cannot compute the posterior pdf/pmf exactly, and we will resort to variational inference to approximate them – an approach often referred to as Variational Continual Learning (VCL) Nguyen et al. (2018). As we shall see in Appendix E, Assumption 1 is needed in VCL to avoid catastrophic forgetting.

## A.1 Relationship between IBCL and other BCL techniques

Like Farquhar and Gal (2019); Li et al. (2020), the weights in our Bayesian neural networks (BNNs) have Gaussian distribution with diagonal covariance matrix. Because IBCL is rooted in Bayesian continual learning, we can initialize IBCL with a much smaller number of parameters to solve a complex task as long as it can solve a set of simpler tasks. In addition, IBCL does not need to evaluate the importance of parameters by measures such as computing the Fisher information, which are computationally expensive and intractable in large models.

### A.1.1 Relationship between IBCL and MAML

In this section, we discuss the relationship between IBCL and the Model-Agnostic Meta-Learning (MAML) and Bayesian MAML (BMAML) procedures introduced in Finn et al. (2017); Yoon et al. (2018b), respectively. These are inherently different than IBCL, since the latter is a continual learning procedure, while MAML and BMAML are meta-learning algorithms. Nevertheless, given the popularity of these procedures, we feel that relating IBCL to them would be useful to draw some insights on IBCL itself.

In MAML and BMAML, a task $i$ is specified by a $n_i$-shot dataset $D_i$ that consists of a small number of training examples, e.g. observations $(x_{1_i}, y_{1_i}), \dots, (x_{n_i}, y_{n_i})$. Tasks are sampled from a task distribution $\mathbb{T}$ such that the sampled tasks share the statistical regularity of the task distribution. In IBCL, Assumption 1 guarantees that the tasks $p_i$ share the statistical regularity of class $\mathcal{F}$. MAML and BMAML leverage this regularity to improve the learning efficiency of subsequent tasks.

At each meta-iteration $i$,

1. *Task-Sampling*: For both MAML and BMAML, a mini-batch $T_i$ of tasks is sampled from the task distribution $\mathbb{T}$. Each task $\tau_i \in T_i$ provides task-train and task-validation data, $D_{\tau_i}^{\text{trn}}$ and $D_{\tau_i}^{\text{val}}$, respectively.

2. *Inner-Update*: For MAML, the parameter of each task $\tau_i \in T_i$ is updated starting from the current generic initial parameter $\theta_0$, and then performing $n_i$ gradient descent steps on the task-train loss. For BMAML, the posterior $q(\theta_{\tau_i} \mid D_{\tau_i}^{\text{trn}}, \theta_0)$ is computed, for all $\tau_i \in T_i$.

3. *Outer-Update*: For MAML, the generic initial parameter $\theta_0$ is updated by gradient descent. For BMAML, it is updated using the Chaser loss (Yoon et al., 2018b, Equation (7)).

Notice how in our work $\bar{w}$ is a probability vector. This implies that if we fix a number of task $k$ and we let $\bar{w}$ be equal to $(w_1, \dots, w_k)^\top$, then $\bar{w} \cdot \bar{p}$ can be seen as a sample from $\mathbb{T}$ such that $\mathbb{T}(p_i) = w_i$, for all $i \in \{1, \dots, k\}$.

Here lies the main difference between IBCL and BMAML. In the latter the information provided by the tasks is used to obtain a refinement of the (parameter of the) distribution $\mathbb{T}$ on the tasks themselves. In IBCL, instead, we are interested in the optimal parameterization of the posterior distribution associated with $\bar{w} \cdot \bar{p}$. Notice also that at time $k + 1$, in IBCL the support of $\mathbb{T}$ changes: it is $\{p_1, \dots, p_{k+1}\}$, while for MAML and BMAML it stays the same.

Also, MAML and BMAML can be seen as ensemble methods, since they use different values (MAML) or different distributions (BMAML) to perform the Outer-Update and come up with a

single value (MAML) or a single distributions (BMAML). Instead, IBCL keeps distributions separate via FGCS, thus capturing the ambiguity faced by the designer during the analysis.

Furthermore, we want to point out how while for BMAML the tasks $\tau_i$ are all "candidates" for the true data generating process (dgp) $p_i$, in IBCL we approximate the pdf/pmf of $p_i$ with the product $\prod_{h=1}^{i} \ell_h$ of the likelihoods up to task $i$. The idea of different candidates for the true dgp is beneficial for IBCL as well: in the future, we plan to let go of Assumption 1 and let each $p_i$ belong to a credal set $\mathcal{P}_i$. This would capture the epistemic uncertainty faced by the agent on the true dgp.

To summarize, IBCL is a continual learning technique whose aim is to find the correct parameterization of the posterior associated with $\bar{w} \cdot \bar{p}$. Here, $\bar{w}$ expresses the developer's preferences on the tasks. MAML and BMAML, instead, are meta-learning algorithms whose main concern is to refine the distribution $\mathbb{T}$ from which the tasks are sampled. While IBCL is able to capture the preferences of, and the ambiguity faced by, the designer, MAML and BMAML are unable to do so. On the contrary, these latter seem better suited to solve meta-learning problems. An interesting future research direction is to come up with imprecise BMAML, or IBMAML, where a credal set $\text{Conv}(\{\mathbb{T}_1, \ldots, \mathbb{T}_k\})$ is used to capture the ambiguity faced by the developer in specifying the correct distribution on the possible tasks. The process of selecting one element from such credal set may lead to computational gains.

## B HIGHEST DENSITY REGION

Equivalently to Definition 2, an HDR is defined as follows (Hyndman, 1996).

**Definition 6.** Let $\Theta$ be a set of interest, and consider a significance level $\alpha \in [0, 1]$. Suppose that a (continuous) random variable $\theta \in \Theta$ has probability density function (pdf) $q$.[4] The $\alpha$-level Highest Density Region (HDR) $\Theta_q^\alpha$ is the subset of $\Theta$ such that

$$\Theta_q^\alpha = \{\theta \in \Theta : q(\theta) \geq q^\alpha\}, \tag{2}$$

where $q^\alpha$ is a constant value. In particular, $q^\alpha$ is the largest constant such that $\text{Pr}_{\theta \sim q}[\theta \in \Theta_q^\alpha] \geq 1 - \alpha$.

Some scholars indicate HDRs as the Bayesian counterpart to the frequentist concept of confidence intervals. In dimension 1, $\Theta_q^\alpha$ can be interpreted as the narrowest interval – or union of intervals – in which the value of the (true) parameter falls with probability of at least $1 - \alpha$ according to distribution $q$. We give a simple visual example in Figure 7.

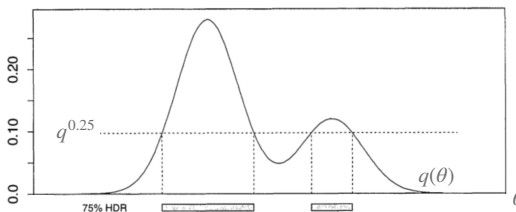

Figure 7: The 0.25-HDR for a Normal Mixture density. This picture is a replica of (Hyndman, 1996, Figure 1). The geometric representation of "75% probability according to $q$" is the area between the pdf curve $q(\theta)$ and the horizontal bar corresponding to $q^{0.25}$. A higher probability coverage (according to $q$) would correspond to a lower constant, so $q^\alpha < q^{0.25}$, for all $\alpha < 0.25$. In the limit, we recover 100% coverage at $q^0 = 0$.

## C ADDITIONAL RELATED WORK

**Multi-task Learning under Preferences.** Learning for Pareto-optimal models under task performance trade-offs has been studied by researchers in multi-task learning (Caruana, 1997; Sener and Koltun, 2018). Various techniques have been applied to obtain models that address particular trade-off points (Lin et al., 2019; 2020; Ma et al., 2020; Gupta et al., 2021). The idea of preferences on

---

[4]Here too, for ease of notation, we do not distinguish between a random variable and its realization.

the trade-off points is introduced in multi-objective optimization (Lin et al., 2020; Sener and Koltun, 2018), and a preference can guide learning algorithms to search for a particular model. We borrow the formalization of preferences from Mahapatra and Rajan (2020), where a preference is given by a vector of non-negative real weights $\bar{w}$, with each entry $w_i$ corresponding to task $i$. That is, $w_i \geq w_j \iff i \succeq j$. This means that if $w_i \geq w_j$, then task $i$ is preferred to task $j$. However, state-of-the-art algorithms require training one model per preference, imposing large overhead when there is a large number of preferences.

**Continual Learning.** Continual learning, also known as lifelong learning, is a special case of multi-task learning, where tasks arrive sequentially instead of simultaneously (Thrun, 1998; Ruvolo and Eaton, 2013a; Silver et al., 2013; Chen and Liu, 2016). In this paper, we leverage Bayesian inference in the knowledge base update (Ebrahimi et al., 2019; Farquhar and Gal, 2019; Kao et al., 2021). Like generic multi-task learning, continual learning also faces the stability-plasticity trade-off (De Lange et al., 2021; Kim et al., 2023; Raghavan and Balaprakash, 2021), which balances between performance on new tasks and resistance to catastrophic forgetting (Kirkpatrick et al., 2017b; Lee et al., 2017; Robins, 1995). Current methods identify models to address trade-off preferences by techniques such as loss regularization (Servia-Rodriguez et al., 2021), meaning at least one model needs to be trained per preference.

Researchers in CL have proposed various approaches to retain knowledge while updating a model on new tasks. These include modified loss landscapes for optimization (Farajtabar et al., 2020; Kirkpatrick et al., 2017a; Riemer et al., 2019; Suteu and Guo, 2019; Tang et al., 2021), preservation of critical pathways via attention (Abati et al., 2020; Serra et al., 2018; Xu et al., 2021; Yoon et al., 2020), memory-based methods (Lopez-Paz and Ranzato, 2017; Rolnick et al., 2019), shared representations (He et al., 2018; Lee et al., 2019; Lu et al., 2017; Ruvolo and Eaton, 2013b; Vandenhende et al., 2019; Yoon et al., 2018a), and dynamic representations (Bulat et al., 2020; Mendez and Eaton, 2021; Ramesh and Chaudhari, 2022; Rusu et al., 2016; Schwarz et al., 2018; Yang and Hospedales, 2017). Bayesian, or probabilistic methods such as variational inference are also adopted (Ebrahimi et al., 2019; Farquhar and Gal, 2019; Kao et al., 2021; Kessler et al., 2023; Li et al., 2020; Nguyen et al., 2018).

## D   2-WASSERSTEIN METRIC

In the main portion of the paper, we endowed $\Delta_{\mathcal{XY}}$ with the 2-Wasserstein metric. It is defined as

$$\|p - p'\|_{W_2} \equiv W_2(p, p') := \sqrt{\inf_{\gamma \in \Gamma(p,p')} \mathbb{E}_{((x_1,y_1),(x_2,y_2)) \sim \gamma}[d((x_1, y_1), (x_2, y_2))^2]}, \quad \text{where}$$

1. $p, p' \in \Delta_{\mathcal{XY}}$;
2. $\Gamma(p, p')$ is the set of all couplings of $p$ and $p'$. A coupling $\gamma$ is a joint probability measure on $(\mathcal{X} \times \mathcal{Y}) \times (\mathcal{X} \times \mathcal{Y})$ whose marginals are $p$ and $p'$ on the first and second factors, respectively;
3. $d$ is the product metric endowed to $\mathcal{X} \times \mathcal{Y}$ (Deza and Deza, 2013, Section 4.2).[5]

We choose the 2-Wasserstein distance for the ease of computation. In practice, when all distributions are modeled by Bayesian neural networks with independent Gaussian weights and biases, we have

$$\|q_1 - q_2\|_{W_2}^2 = \|\mu_{q_1}^2 - \mu_{q_2}^2\|_2^2 + \|\sigma_{q_1}^2 \mathbf{1} - \sigma_{q_2}^2 \mathbf{1}\|_2^2, \tag{3}$$

where $\|\cdot\|_2$ denotes the Euclidean norm, $\mathbf{1}$ is a vector of all 1's, and $\mu_q$ and $\sigma_q$ are respectively the mean and standard deviation of a multivariate normal distribution $q$ with independent dimensions, $q = \mathcal{N}(\mu_q, \sigma_q^2 I)$, $I$ being the identity matrix. Therefore, computing the $W_2$-distance between two distributions is equivalent to computing the difference between their means and variances.

## E   IMPORTANCE OF ASSUMPTION 1

We need Assumption 1 in light of the results in Kessler et al. (2023). There, the authors show that misspecified models can forget even when Bayesian inference is carried out exactly. By requiring

---

[5]We denote by $d_{\mathcal{X}}$ and $d_{\mathcal{Y}}$ the metrics endowed to $\mathcal{X}$ and $\mathcal{Y}$, respectively.

that $\text{diam}(\mathcal{F}) = r$, we control the amount of misspecification via $r$. In Kessler et al. (2023), the authors design a new approach – called Prototypical Bayesian Continual Learning, or ProtoCL – that allows dropping Assumption 1 while retaining the Bayesian benefit of remembering previous tasks. Because the main goal of this paper is to come up with a procedure that allows the designer to express preferences over the tasks, we retain Assumption 1, and we work in the classical framework of Bayesian Continual Learning. In the future, we plan to generalize our results by operating with ProtoCL.[6]

## F    AN EXAMPLE OF A PARAMETERIZED FAMILY $\mathcal{F}$

Let us give an example of a parameterized family $\mathcal{F}$. Suppose that we have one-dimensional data points and labels. At each task $i$, the marginal on $\mathcal{X}$ of $p_i$ is a Normal $\mathcal{N}(\mu, 1)$, while the conditional distribution of label $y \in \mathcal{Y}$ given data point $x \in \mathcal{X}$ is a categorical $\text{Cat}(\vartheta)$. Hence, the parameter for $p_i$ is $\theta = (\mu, \vartheta)$, and it belongs to $\Theta = \mathbb{R} \times \mathbb{R}^{|\mathcal{Y}|}$. In this situation, an example of a family $\mathcal{F}$ satisfying Assumptions 1 and 2 is the convex hull of distributions that can be decomposed as we just described, and whose distance according to the 2-Wasserstein metric does not exceed some $r > 0$.

## G    PROOFS OF THE THEOREMS

*Proof of Theorem 4.* Without loss of generality, suppose we have encountered $i = 2$ tasks so far, so the FGCS is $\mathcal{Q}_2$. Let $\text{ex}[\mathcal{Q}_1] = \{q_1^j\}_{j=1}^{m_1}$ and $\text{ex}[\mathcal{Q}_2] \setminus \text{ex}[\mathcal{Q}_1] = \{q_2^j\}_{j=1}^{m_2}$. Let $\hat{q}$ be any element of $\mathcal{Q}_2$. Then, there exists a probability vector $\bar{\beta} = (\beta_1^1, \ldots, \beta_1^{m_1}, \beta_2^1, \ldots, \beta_2^{m_2})^\top$ such that

$$\hat{q} = \sum_{j=1}^{m_1} \beta_1^j q_1^j + \sum_{j=1}^{m_2} \beta_2^j q_2^j \propto \hat{p}_1 \sum_{j=1}^{m_1} \beta_1^j q_0^j + \hat{p}_2 \sum_{j=1}^{m_2} \beta_2^j q_0^j. \tag{4}$$

Here, $\hat{p}_i = \prod_{k=1}^{i} \ell_k$, and $\ell_k$ is the likelihood at task $k$. It estimates the pdf of the true data generating process $p_i$ of task $i$. The proportional relationship in equation 4 is based on the Bayesian inference step (line 3, approximated via variational inference) of Algorithm 1. We can then find a vector $\bar{w} = (w_1 = \sum_{j=1}^{m_1} \beta_1^j, w_2 = \sum_{j=1}^{m_2} \beta_2^j)^\top$ that expresses the designer's preferences over tasks 1 and 2. As we can see, then, the act of selecting a generic distribution $\hat{q} \in \mathcal{Q}_2$ is equivalent to specifying a preference vector $\bar{w}$ over tasks 1 and 2. This concludes the proof. □

*Proof of Theorem 5.* For maximum generality, assume $\Theta$ is uncountable. Recall from Definition 2 that $\alpha$-level Highest Density Region $\Theta_{\bar{w}}^\alpha$ is defined as the subset of the parameter space $\Theta$ such that

$$\int_{\Theta_{\bar{w}}^\alpha} \hat{q}_{\bar{w}}(\theta) \mathrm{d}\theta \geq 1 - \alpha \quad \text{and} \quad \int_{\Theta_{\bar{w}}^\alpha} \mathrm{d}\theta \text{ is a minimum.}$$

We need $\int_{\Theta_{\bar{w}}^\alpha} \mathrm{d}\theta$ to be a minimum because we want $\Theta_{\bar{w}}^\alpha$ to be the smallest possible region that gives us the desired probabilistic coverage. Equivalently, from Definition 6 we can write that $\Theta_{\bar{w}}^\alpha = \{\theta \in \Theta : \hat{q}_{\bar{w}}(\theta) \geq \hat{q}_{\bar{w}}^\alpha\}$, where $\hat{q}_{\bar{w}}^\alpha$ is the largest constant such that $\text{Pr}_{\theta \sim \hat{q}_{\bar{w}}}[\theta \in \Theta_{\bar{w}}^\alpha] \geq 1 - \alpha$. Our result $\text{Pr}_{\theta_{\bar{w}}^\star \sim \hat{q}_{\bar{w}}}[\theta_{\bar{w}}^\star \in \Theta_{\bar{w}}^\alpha] \geq 1 - \alpha$, then, comes from the fact that $\text{Pr}_{\theta_{\bar{w}}^\star \sim \hat{q}_{\bar{w}}}[\theta_{\bar{w}}^\star \in \Theta_{\bar{w}}^\alpha] = \int_{\Theta_{\bar{w}}^\alpha} \hat{q}_{\bar{w}}(\theta) \mathrm{d}\theta$, a consequence of a well-known equality in probability theory (Billingsley, 1986). □

## H    DETAILS OF EXPERIMENTS

### H.1    SETUP DETAILS

We select 15 tasks from CelebA. All tasks are binary image classification on celebrity face images. Each task $i$ is to classify whether the face has an attribute such as wearing eyeglasses or having a mustache. The first 15 attributes (out of 40) in the attribute list (Liu et al., 2015) are selected for our

---

[6]In Kessler et al. (2023), the authors also show that if there is a task dataset imbalance, then the model can forget under certain assumptions. To avoid complications, in this work we tacitly assume that task datasets are balanced.

tasks. The training, validation and testing sets are already split upon download, with 162,770, 19,867 and 19,962 images, respectively. All images are annotated with binary labels of the 15 attributes in our tasks. We use the same training, validation and testing set for all tasks, with labels being the only difference.

We select 20 classes from CIFAR100 (Krizhevsky et al., 2009) to construct 10 Split-CIFAR100 tasks (Zenke et al., 2017). Each task is a binary image classification between an animal class (label 0) and a non-animal class (label 1). The classes are (in order of tasks):

1. Label 0: aquarium fish, beaver, dolphin, flatfish, otter, ray, seal, shark, trout, whale.
2. Label 1: bicycle, bus, lawn mower, motorcycle, pickup truck, rocket, streetcar, tank, tractor, train.

That is, the first task is to classify between aquarium fish images and bicycle images, and so on. We want to show that the continual learning model incrementally gains knowledge of how to identify animals from non-animals throughout the task sequence. For each class, CIFAR100 has 500 training data points and 100 testing data points. We hold out 100 training data points for validation. Therefore, at each task we have $400 \times 2 = 800$ training data, $100 \times 2 = 200$ validation data and $100 \times 2 = 200$ testing data.

We also select 20 classes from TinyImageNet (Le and Yang, 2015). The setup is similar to Split-CIFAR100, with label 0 being animals and 1 being non-animals.

1. Label 0: goldfish, European fire salamander, bullfrog, tailed frog, American alligator, boa constrictor, goose, koala, king penguin, albatross.
2. Label 1: cliff, espresso, potpie, pizza, meatloaf, banana, orange, water tower, via duct, tractor.

The dataset already splits 500, 50 and 50 images for training, validation and testing per class. Therefore, each task has 1000, 100 and 100 images for training, validation and testing, respectively.

20NewsGroups (Lang, 1995) contains news report texts on 20 topics. We select 10 topics for 5 binary text classification tasks. Each task is to distinguish whether the topic is computer-related (label 0) or not computer-related (label 1), as follows.

1. Label 0: comp.graphics, comp.os.ms-windows.misc, comp.sys.ibm.pc.hardware, comp.sys.mac.hardware, comp.windows.x.
2. Label 1: misc.forsale, rec.autos, rec.motorcycles, rec.sport.baseball, rec.sport.hockey.

Each class has different number of news reports. On average, a class has 565 reports for training and 376 for testing. We then hold out 100 reports from the 565 for validation. Therefore, each binary classification task has 930, 200 and 752 data points for training, validation and testing, on average respectively.

All data points are first preprocessed by a feature extractor. For images, the feature extractor is a pre-trained ResNet18 (He et al., 2016). We input the images into the ResNet18 model and obtain its last hidden layer's activations, which has a dimension of 512. For texts, the extractor is TF-IDF (Aizawa, 2003) succeeded with PCA to reduce the dimension to 512 as well.

Each Bayesian network model is trained with evidence lower bound (ELBO) loss, with a fixed feed-forward architecture (input=512, hidden=64, output=1). The hidden layer is ReLU-activated and the output layer is sigmoid-activated. Therefore, our parameter space $\Theta$ is the set of all values that can be taken by this network's weights and biases.

The three variational inference priors, learning rate, batch size and number of epcohs are tuned on validation sets. The tuning results are as follows.

1. CelebA: priors = $\{\mathcal{N}(0, 0.2^2 I), \mathcal{N}(0, 0.25^2 I), \mathcal{N}(0, 0.3^2 I)\}$, lr = $1e-3$, batch size = 64, epochs = 10.
2. Split-CIFAR100: priors = $\{\mathcal{N}(0, 2^2 I), \mathcal{N}(0, 2.5^2 I), \mathcal{N}(0, 3^2 I)\}$, lr = $5e-4$, batch size = 32, epochs = 50.
3. TinyImageNet: priors = $\{\mathcal{N}(0, 2^2 I), \mathcal{N}(0, 2.5^2 I), \mathcal{N}(0, 3^2 I)\}$, lr = $5e-4$, batch size = 32, epochs = 30.

4. 20NewsGroup: priors = $\{\mathcal{N}(0, 2^2 I), \mathcal{N}(0, 2.5^2 I), \mathcal{N}(0, 3^2 I)\}$, lr = $5e - 4$, batch size = 32, epochs = 100.

For the baseline methods, we use exactly the same learning rate, batch sizes and epochs. For probabilistic baseline methods (VCL), we use the prior with the median standard deviation. For example, on CelebA tasks, VCL uses the normal prior $\mathcal{N}(0, 0.25^2 I)$.

For rehearsal-based baselines, the memory size per task for CelebA is 200, and for the rest is 50. Together with the numbers above, we can compute the numerical values in Table 1.

## H.2 ADDITIONAL RESULTS

In Figure 8 and 9, we provide visual representations of the performances of IBCL vs. the baseline methods on Split CIFAR-100 and CelebA, respectively. So far, to our knowledge, there is no discussion on how to specify a task trade-off preference in prompt-based continual learning, and we make an attempt by using a preference-weighted sum of all learned prompts in L2P. We can see how this approach generally works poorly, except for CelebA, where L2P performs nearly well as IBCL. We believe the performance by prompts trained in L2P depends on its frozen model, and how to use prompt-based methods to generate preference-specified models is still an open problem.

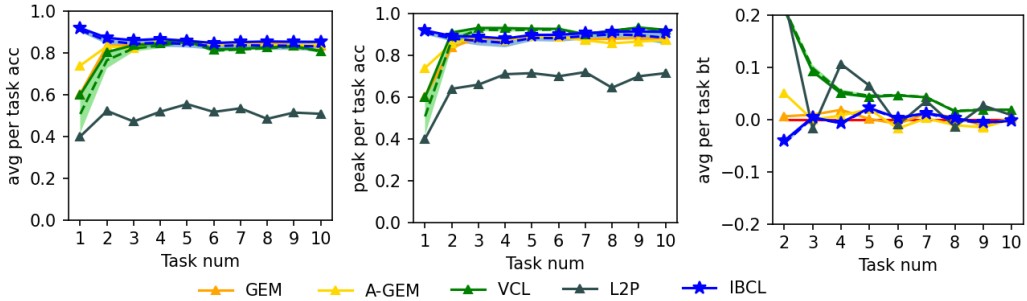

Figure 8: Results of Split-CIFAR-100.

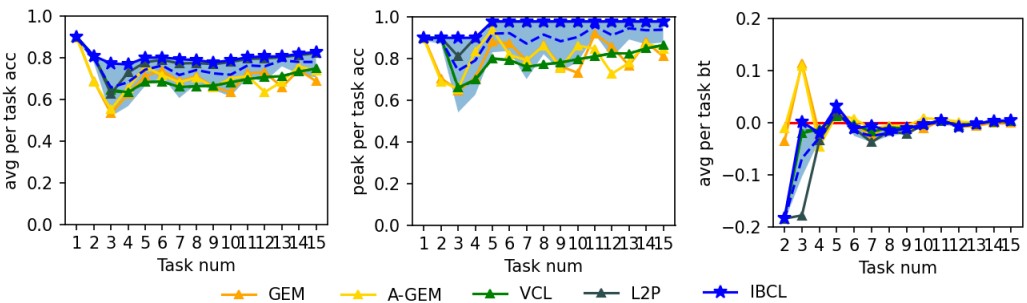

Figure 9: Results of CelebA.

