# OpenReview forum: "IBCL: Zero-shot Model Generation for Task Trade-offs in Continual Learning"
_ICLR.cc/2024/Conference — Submitted to ICLR 2024_

### Official Review · Reviewer_bswF · 2023-10-30

**Soundness:** 3 good
**Presentation:** 3 good
**Contribution:** 3 good
**Rating:** 8
**Confidence:** 4

**Summary:**

This paper proposes Imprecise Bayesian Continual Learning (ICBL) for continual learning under large number of preferences. Unlike other approaches which have to learn a model for each preference, ICBL can handle arbitrary number of preferences under limited compute by updating a knowledge base (aggregated over tasks) as a convex hull over model parameters. As only extreme point of the hull have to be stored, this can be done with lower memory requirements. Leveraging this knowledge base, preference-addressing models can be generated without training, and have guarantees w.r.t. Pareto optimal parameters.
Results on four image classification and NLP benchmarks show convincing results with effectively no negative backward transfer.

**Strengths:**

* To the best of my knowledge, the proposed method - ICBL - is the first to tackle the task of continual learning with a large number of preferences.
* Its derivation and motivation appears sensible, and can account for an arbitrary number of preferences without costly retraining.

**Weaknesses:**

In this section, I include both issues I have with the paper, and general questions regarding my understanding of the proposed approach.

__Weaknesses__

* The paper itself is too dense - a lot of the crucial intuition and motivation is moved to the appendix, which makes the main paper very difficult to parse, for example
	* Separation and placement in literature of the proposed method IBCL and MAML/BMAML (App. A).
	* The reason for working in a bayesian CL setting (App. B)
	* The importance of the particular task similarity assumption (App. E)
	* Limited discussion on how preferences are formalized (App. G)
	* Very basic experimental details on the CL experiments (App. J)
Without continuously looking at the supplementary, understanding both key elements of the method and its motivation become in parts near impossible. This specifically refers to the reason behind bayesian CL, and details regarding the made assumptions. It would be great to see that changed.

* The proposed setup seems contrived - in particular the continual aspect, with both examples provided in the introduction primarily highlighting the multi-task nature of the problem. The continual aspect seems to be mostly a sidenote (e.g. "preference may even change over time"). Similarly, the authors only provide limited references for the relevance of the described problem scenario where a large number of preferences has to be accounted for continually over time. It would be great if the authors could offer some more clarity here.

* I may be missing something here, but the authors list poorly performing models to also be sampled from the HDRs, and require separate evaluation on a withheld validation set. This seems like a rather crucial point to elaborate on - what exactly is the rate of poorly performing models, and how dependent is ICBL on selection using validation metrics?

* The experimental studies are limited, and only compares to a single continual learning method, while not providing any context as to why GEM was selected in particular. Are other methods not suitable for this scenario? Similarly, can ICBL not be deployed on standard, single-preference continual learning benchmarks? This would be great to understand, and if not, why.


__Questions__


* What drives the definition of the Assumption 1 for Task Similarity? In particular, how close it is to a realistic assumption (in particular F being a convex subset of \Delta_{xy})? Conceptual motivation for such a key assumption would make it easier to grasp the proposed approach quicker.

* Intuitively, it seems like the diameter of the convex subsets F (r) could connect to the expected continuous distribution shifts that can be handled. Is that right? And generally, how does the choice of r drive/change the behaviour/applicability of ICBL?

**Questions:**

I am currently having some trouble correctly placing the relevance of the tackled problem, alongside issues with the experiments and questions regarding the proposed approach - as listed in the above section.
I am happy to raise my score if these can be adequately addressed!

---

> ### Author Response · Authors · 2023-11-17
> **Revised manuscript submitted**
>
> We appreciate your thoughtful feedback, and we invite you to read our revised manuscript, which significantly improves readability and addresses your concerns.
>
> “The paper itself is too dense.”
>
> We have improved readability in our revised manuscript. Please also refer to our comment to everyone for a list of our modifications.
>
> “The proposed setup seems contrived.”
>
> Our setup is simplified and clarified in the revised manuscript. Please read our revised version of the Introduction for a better structured story.
>
> “Poorly performing models can also be sampled from HDRs.”
>
> We are able to completely avoid sampling these models by modifying the hyperparameter alpha. Please refer to our ablation studies in Appendix H.3 of the revised manuscript, where we are able to always sample near Pareto-optimal models with a small alpha (0.01). In fact, avoiding poor models does not depend on validation at all, and we made this correction in our latest manuscript.
>
> “The experimental studies are limited.”
>
> We have added experiments based on the suggestions of reviewer 79Li. As in our revised manuscript, GEM and A-GEM are selected because they are rehearsal-based, and there is an established method to specify task preferences on rehearsal-based continual learning – that is, using preferences as weights on loss functions. The other methods, such as prompt-based continual learning, have no standard way to specify a task preference so far. In our revised experiments, we made an attempt to use a preference-weighted sum of prompts in L2P, but it ended up with poor performance except for the CelebA benchmark.
>
> “Can IBCL be deployed to single-preference benchmarks?”
>
> Yes. We can request IBCL to generate models for any number of preferences (including one preference only), and the total training overhead (measured in number of batch updates) remains constant, regardless of the number of preferences requested.
>
> “What drives the assumption of task similarity?”
>
> Task similarity assumption is needed to mitigate the possible model misspecification, which in turn could lead to catastrophic forgetting even when Bayesian inference is carried out exactly. This is now explicitly mentioned after presenting Assumption 1, and discussed in more depth in Appendix E.
>
> “Does the diameter of F(r) connect to the expected continuous distribution shifts that can be handled? How does the choice of r affect the applicability of IBCL?”
>
> Yes: assuming a bounded diameter implies that the tasks are somehow close to each other. In other words, the distribution shift between the distributions for task i and j, p_i and p_j respectively, must not be too pronounced. If we were to allow arbitrary differences between the tasks the agent needs to accomplish, as a consequence of IBCL being embedded in a Bayesian framework, we may incur catastrophic forgetting. As pointed out in Appendix E, in the future, we will generalize IBCL to ProtoCL instead of Bayesian learning to let go of Assumption 1. Having a bounded diameter does not limit the applicability of IBCL, since in the majority of applications the tasks we want to accomplish are reasonably similar to each other. This is for example the case of classification, where Assumption 1 should always be satisfied, with a reasonably small diameter r.
>
> Again, we appreciate your comments and hope the rating can be adjusted accordingly.

---

> > ### Comment · Reviewer_bswF · 2023-11-19
> >
> > I thank the authors for their detailed feedback. In particular, I very much appreciate the motivational changes in the introduction (especially the example), which motivate the problem setting a lot better.
> > This translates to the overall writing of the paper, which has notably improved, and is now much easier to parse and understand:
> > * The change in Fig. 1 is much easier to parse, and aligns better with the motivational example.
> > * The overall method description & motivation reads much better, with Alg.1 now quicker to understand and less cluttered. More importantly however, it is now well integrated into the main text, and each key step is explained. This allowed me to understand the method in more detail - I very much appreciate that.
> >
> > Overall, the paper now is much more self-contained and comprehensively written. As a consequence, I now do believe that the problem setup of continual learning that cheaply accounts for preference changes is a sensible one, and the proposed approach of particular interest to the research community. As such, I have raised my score to 6. I will be happy to raise it to 8 if the remaining issues (see below) are addressed.
> >
> >
> > __Remaining notes on my end:__
> > * The paper now only stands at 7.5 pages - so a lot of context was cut (and moved to the appendix), such as the additional benchmark results. Especially the latter should still be included in the paper, as opposed to results on a single benchmark. With all the positive changes w.r.t. the paper writing, the experimental section still reads as a sidenote unfortunately - an issue exacerbated by the removal of experimental results. These should be included, and discussed in more detail. Similarly, ablation studies on the two key hyperparameters should definitely be incorporated in the main paper, as it is otherwise very difficult to experimental understand hyperparameter choice impact, and the corresponding method dependency.
> > * Could the authors elaborate more on the differences in backward transfer in Fig. 2, especially compared to VCL.
> > * It would be great to reference Alg. 1/2 in Alg. 3.
> > * The Theorem-formatting is off.
> > * I do wish Table and Figure captions (particularly Fig. 2, Tab. 1) were actually self-contained.

---

> > > ### Author Response · Authors · 2023-11-20
> > >
> > > Dear reviewer, thank you so much for your additional thoughtful comments. We have submitted a second revision to address your concerns, including the items as follows.
> > >
> > > 1. The paper is now 9 pages, with additional benchmark results and ablation studies brought from appendix to the main text. More discussion on the results are also written in our new experiment section.
> > >
> > > 2. In our updated Section 5.2, we have discussed the backward transfer pattern. We believe this pattern may be caused by the choice of priors, the choice of variational inference method, or an intrinsic characteristic of IBCL. The cause for such a pattern is indeed a future direction to research on. However, we can still see IBCL's backward transfer quickly returns to near-zero to positive, while VCL's decreases, showing that IBCL does not forget and has similar capability to retain knowledge as VCL in long run.
> > >
> > > 3. Alg 1 and Alg 2 are referred to in Alg 3.
> > >
> > > 4. We assume by "theorem formatting" you mean the indexing of theorems and definitions are off. We now fixed the indexing. Please let us know if this means something else and we are willing to change it.
> > >
> > > 5. The table and figure captions are now self-contained.
> > >
> > > Thanks again for considering raising the rating!

---

> > > > ### Comment · Reviewer_bswF · 2023-11-20
> > > >
> > > > I appreciate the quick changes, and am quite happy with the current version of the paper - the problem is now well motivated, the writing has been improved alongside better visuals (e.g. Fig. 1), the experimental comparisons are more comprehensive and better explained (thereby also better differentiating the proposed IBCL from existing approaches), and with the inclusion of hyperparameter ablations the experiments are more self-contained.
> > > >
> > > > As such, I will raise my score to 8, and opt for acceptance.
> > > >
> > > > Some small notes that I hope will be addressed at least for the camera-ready:
> > > > * By formatting, I mainly referred to a cursive font for the Theorems / Definitions, or generally e.g. some better visual separation from the main text. The indexing seemed fine to me.
> > > > * I appreciate the more comprehensive figure & table captions. It would be nice if this could also be done for the newly added ones.

---

> > > > > ### Author Response · Authors · 2023-11-20
> > > > >
> > > > > Thank you very much for raising the rating. We will address your notes at the camera-ready version.

---

### Official Review · Reviewer_79Li · 2023-10-31

**Soundness:** 2 fair
**Presentation:** 2 fair
**Contribution:** 2 fair
**Rating:** 3
**Confidence:** 4

**Summary:**

They propose Imprecise Bayesian Continual Learning, and the proposed method has the two pros: (1) update the knowledge in the form of a convex of model parameters (2) it does not require additional training cost. Also, they show that models from IBCL obtain pareto optimal parameters.

**Strengths:**

**1 [Motivation].** I agree with the authors' claim and the philosophy of the method seems make sense for me. Especially, reducing the training overhead is very important topic in continual learning area.

**2 [Guarantee pareto optimal].** While I'm not familiar with the mathematical analysis, the authors guarantee that model generation from IBCL has pareto optimal parameters for each task. This work seems impressive for me.

**Weaknesses:**

**1 [Lack of Baselines]. ** In my opinion, the experiment evidence needs to be improved. Especially the baselines are too old and not enough to prove that the proposed method is state-of-the-art. I will propose some recent baselines as below:

**Zero-shot.** Since the proposed method argues the benefit of zeroshot, it needs to compare with the models that have zero-shot capabilities. I recommend CLIP[1] as a baseline, but if the authors think that CLIP is unfair for comparison, it is also fine to compare with traditional zero-shot learning techniques.

[1] Radford, Alec, et al. "Learning transferable visual models from natural language supervision." International conference on machine learning. PMLR, 2021.

** Efficient training method.** There are several continual learning papers with a few training cost. In recent, there are lots of those kinds of papers[2, 3, 4], so it would be good baselines to validate the effectiveness of proposed method.

[2] Wang, Zifeng, et al. "Learning to prompt for continual learning." Proceedings of the IEEE/CVF Conference on Computer Vision and Pattern Recognition. 2022.

[3] Wang, Zifeng, et al. "Dualprompt: Complementary prompting for rehearsal-free continual learning." European Conference on Computer Vision. Cham: Springer Nature Switzerland, 2022.

[4] Smith, James Seale, et al. "CODA-Prompt: COntinual Decomposed Attention-based Prompting for Rehearsal-Free Continual Learning." Proceedings of the IEEE/CVF Conference on Computer Vision and Pattern Recognition. 2023.

**Advanced Research after GEM.** In fact, there are advanced work after publishing GEM. I will share the paper[5]. Since the authors select the GEM as a baseline, it would be better to compare the method with A-GEM too.

[5] Chaudhry, Arslan, et al. "Efficient lifelong learning with a-gem." arXiv preprint arXiv:1812.00420 (2018).

Lastly, I will upgrade my rating if you enhance the experiment part.

**Questions:**

I already wrote my concerns in weakness parts.

---

> ### Author Response · Authors · 2023-11-17
> **New experiments conducted**
>
> We appreciate your thoughtful comments, and we invite you to read our revised manuscript, which significantly improves readability and compares to two more baselines: A-GEM [1] and L2P [2]. Due to time constraints, we are unable to finish working on the other baselines you suggested, but we will keep working on them.
>
> We indeed believe CLIP [3] is a well-established zero-shot approach. However, it requires natural languages as prompts, which is unavailable in our settings. Therefore, we choose L2P, where prompts are trainable parameters.
>
> Our new experiments show that IBCL is able to obtain high continual learning performance, thanks to its probabilistic Pareto-optimality. Moreover, it is able to maintain a constant training overhead per task, regardless of how many preferences are requested. Compared to rehearsal-based baselines, like GEM and A-GEM, the performance is not worsened while the training overhead is significantly reduced.
>
> Although prompt-based baselines, like L2P, also achieves constant training overhead per task, IBCL has a much higher performance overall. This is due to the fact that there is no standard way of specifying preferences over tasks to prompt-based methods, and we made an attempt to train one prompt per task, and use a preference-weighted sum of the learned prompts to specify a preference. The resulting performance is poor except for on the CelebA benchmark. As a consequence, we point out how specifying task preferences in prompt-based continual learning can be a future research direction.
>
> Again, we appreciate your comments that helped us improve the overall quality of the manuscript, and we sincerely hope you can update the rating accordingly.
>
> [1] Chaudhry, Arslan, et al. "Efficient lifelong learning with a-gem." arXiv preprint arXiv:1812.00420 (2018).
>
> [2] Wang, Zifeng, et al. "Learning to prompt for continual learning." Proceedings of the IEEE/CVF Conference on Computer Vision and Pattern Recognition. 2022.
>
> [3] Radford, Alec, et al. "Learning transferable visual models from natural language supervision." International conference on machine learning. PMLR, 2021.

---

> > ### Author Response · Authors · 2023-11-21
> > **Less than 48 hours remained in rebuttal period**
> >
> > Dear reviewer 79Li, it will be much appreciated if you can consider taking a look at our new experiments per your comments. Moreover, the current revised manuscript is much more readable. As the other reviewers are giving positive ratings, we will be grateful if you can consider raising your rating according to our changes. Thank you in advance.

---

### Official Review · Reviewer_SCN6 · 2023-11-20

**Soundness:** 4 excellent
**Presentation:** 3 good
**Contribution:** 3 good
**Rating:** 6
**Confidence:** 4

**Summary:**

This paper successfully achieves the goal of developing classification models for domain-incremental learning, considering user preferences for task trade-offs. Moreover, the learned model is efficient and guarantees Pareto-optimality. The results substantiate the claim that IBCL not only attains high performance through probabilistic Pareto optimality but also excels in the efficient, zero-shot generation of models.

**Strengths:**

For the current revision:

- The paper is well-written and easy to follow, with clear logic throughout. The authors have effectively used bullet points to delineate their settings and motivations, providing a lucid understanding of their objectives.
- The proposed new setting of training Pareto-optimal models under user trade-off preferences between tasks is both significant and well addressed in this context.
- The theoretical framework is self-contained, and the experimental comparisons are comprehensive.

**Weaknesses:**

- The HDR concept (highlighted in purple in Fig.1) is not immediately clear. It would be more helpful to use its full name, 'high density region'. The term 'finitely generated credal set'  presents a similar issue, needing a more explicit definition.
- While the paper mentions several preference-conditioned Pareto models [1, 2] in the appendix, a more detailed explanation of how IBCL differs from these models would be beneficial.

[1] Learning the Pareto Front with Hypernetworks.

[2] Controllable Pareto Multi-task Learning.

**Questions:**

In the experiments, it appears that a smaller $\alpha$ value is preferable. Why not choose an even smaller $\alpha$ (e.g., 0.001)?

---

> ### Author Response · Authors · 2023-11-20
> **Further revised manuscript submitted**
>
> Dear reviewer, thank you for your positive rating and thoughtful comments. We have updated a further revised manuscript that addresses your concerns.
>
> “The HDR concept is not immediately clear.”
>
> We modified Figure 1 as well as the paragraph right after it. In Figure 1, the full names of FGCS and HDR are provided. In the paragraph, we give a more detailed description about what FGCS and HDR are. We also added a reference to Section 2 at the end of the paragraph, to show where formal definitions can be found.
>
> “A more detailed explanation of how IBCL differed from [1] [2] would be beneficial.”
>
> We detailed this explanation in the last two paragraphs before “Contributions” in the Introduction. That is, compared to existing efficient knowledge transfer methods like [1], IBCL enables learning models under specified task trade-off preferences. Compared to existing methods that enable specified preferences like [2], IBCL is not only much more efficient by maintaining a constant training overhead per task, but also guarantees probabilistic Pareto optimality. Moreover, IBCL is a Bayesian CL method, hence it mitigates catastrophic forgetting by design. We prove this and discuss Bayesian CL at length in Appendix A.
>
> “Why not choose a smaller alpha, e.g. 0.001?”
>
> Indeed, a smaller alpha is more preferable. However, when alpha becomes very small, like 0.01, we already have a very high probability of sampling the Pareto-optimal model, and this probability can be hardly improved by using an even smaller alpha. For example, in Figure 2, the shaded blue area that illustrates IBCL’s performance range has already converged to a single curve. This means that the worst sampled model and the best sampled model (Pareto-optimal model) have a negligible difference. Therefore, there is no need to keep shrinking down this difference by using a smaller alpha.
>
> We sincerely hope that you can consider improving the rating based on the changes. Thank you in advance.
>
> [1] Learning the Pareto Front with Hypernetworks.
>
> [2] Controllable Pareto Multi-task Learning.

---

### Official Review · Reviewer_6n6n · 2023-11-22

**Soundness:** 2 fair
**Presentation:** 3 good
**Contribution:** 2 fair
**Rating:** 3
**Confidence:** 4

**Summary:**

The paper proposes a continual learning model, Imprecise Bayesian Continual Learning (IBCL), which accepts the user preference and generates the user-specific model without any training. The IBCL updates a knowledge base in the form of a convex hull of model parameter distributions. The proposed approach also ensures that the buffer growth is sublinear with the increase of tasks. The paper proposes FGCS knowledge base update and HDR computations, which, in certain constraints, help to obtain Probabilistic Pareto-optimality. The results and ablation are shown on the 20NewsGroup datasets. Also, the model requires fewer batch updates at the last task in comparison to its competitor.

**Strengths:**

1. The idea to generate the model without training on the fly, given the user preference, is interesting; it may have wide use for the various problems.
2. The paper provides the theoretical guarantee, but it is not clear how Pareto-optimality helps to improve the model performance.
3. The ablations are convincing.

**Weaknesses:**

1. In Algorithm-1 paper shows the FGCS Knowledge Base Update, which is based on some distance, mostly selecting the samples that have maximum diversity. There are many similar works based on entropy, loss, and other metrics (please refer to [a]).
2. The paper is motivated as we have a large number of users, and the model is scalable for the larger number, but the results are shown only for the 5/10 task, which is small and does not align with the motivation.
3. The baseline papers are outdated; the recent work shows much better results even without replay samples.
4. The ablations are convincing, but the results are insufficient to evaluate the model. The used datasets are limited, and the training procedure is not clear.

Reference: \
[a]  Streaming LifeLong Learning With Any-Time Inference, ICRA-2023

Justification for the rating: The claim in the motivation is not supported in the experiments, lack of motivation about Pareto-optimality, weak/unfair baselines, and the computation of the HDR are not discussed, which is key in the paper.

**Questions:**

1. In Algorithm-1 paper shows the FGCS Knowledge Base Update, which is based on some distance, mostly selecting the samples that have maximum diversity. There are many similar works based on entropy, loss, and other metrics (please refer to [b]). What advantages do they have over the other? Most of the earlier work used fixed/constant size buffers, which is better than sublinear growth. When the results are evaluated, IBCL has sublinear growth; however, the compared method GEM/A-GEM uses a fixed-size buffer. In this scenario, how do the authors ensure a fair comparison? Also, the L2P is a replay-free model, which is not a fair comparison since the model used the replay buffer. What is the buffer growth rate, and how does the performance change with the sublinear growth?
2. The baseline papers are outdated; the recent work shows much better results even without replay samples. Please include the recent replay-based model in the baseline. Also, the L2P (is only a recent model) is a replay-free prompting-based model and there are many updated prompting-based approaches [a, c], etc. which should be included in the baseline.
3. The motivation behind the probabilistic Pareto-optimality is not clear. Why is it important for continual learning?
4. There are no clear descriptions about the HDR computation, i.e., how Algo-2, line-5 computes the HDR? It looks like Preference HDR Computation is expensive, and as the task grows, the complexity increases. Please discuss its computation method and complexity.
5. The ablations are convincing, but the results are insufficient to evaluate the model. The used datasets are limited, and the training procedure is not clear.

Reference: \
[a] CODA-Prompt: Continual Decomposed Attention-based Prompting for Rehearsal-Free Continual Learning, CVPR-2023 \
[b] Streaming LifeLong Learning With Any-Time Inference, ICRA-2023 \
[c] DualPrompt: Complementary Prompting for Rehearsal-free Continual Learning, ECCV-2022

---

> ### Author Response · Authors · 2023-11-22
>
> Thank you for your comments. We believe there are some misunderstandings in our work, and we are happy to address them here. We would also appreciate the reviewer to refer to the comments and our discussions with the other three reviewers. First, for the weaknesses:
>
> **1. “In Algorithm-1 paper shows the FGCS Knowledge Base Update, which is based on some distance … There are many similar works based on entropy, loss, and other metrics.”**
>
> This distance-based distribution selection is not the main focus of this paper. We use this as a convenient subroutine to exclude distributions that are very similar to what is already cached, so that we have a sublinear buffer growth. Our main contribution is modeling the representation learned in continual learning as an FGCS, which allows zero-shot identification of probabilistically Pareto-optimal models.
>
> **2. “The paper is motivated as we have a large number of users, and the model is scalable for the larger number, but the results are shown only for the 5/10 task, which is small and does not align with the motivation.”**
>
> The number of tasks and the number of preferences are two different concepts. We can have a small number of tasks, but with each task we have a large number of preferences. These two numbers are distinguished by $n_{pref}$ and $i$ in our Table 1.
>
> **3. “The baseline papers are outdated.”**
>
> We have explained the baseline selection in Section 5.1, as well as an explanation in the last paragraph before Figure 1 in Section 1 (“However, we identify two major disadvantages …”). That is, so far, people can only specify preferences over tasks on rehearsal-based continual learning algorithms, by using a preference-weighted sum on the losses of the rehearsal memory, making them comparable to IBCL. The "new" and “efficient” algorithms, such as model-based or prompt-based algorithms, do not enable an input of preferences. However, we still made an attempt to input preferences to L2P after the discussion with reviewer 79Li. This is a relatively new prompt-based method for comparison.
>
> **4. “The ablation results are insufficient to evaluate the model.”**
>
> We would appreciate the reviewer to point out how to make the ablation studies sufficient. We conduct these ablations by using different hyperparameters $d$ and $\alpha$ and the results already show a clear trend of how the performance can be affected. The training procedure is exactly the same as in the main experiments, which is detailed in Section 5.1 and Appendix H, just with different hyperparameters given.
>
> Then, for the justification in weaknesses:
>
> **5. “The claim in the motivation is not supported in the experiments.”**
>
> Our experiments already show that IBCL can achieve high continual learning performance metrics, supporting the claim of probabilistic Pareto-optimality. We have also analyzed its overhead in Table 1, supporting the claim of zero-shot model generation under preferences. We are not sure why the reviewer thinks the claim is not supported.
>
> **6. “ Lack of motivation about Pareto-optimality.”**
>
> Pareto-optimality is a common goal in multi-objective optimization, which is the nature of multi-task and continual learning. This motivation is already well supported by the cited previous work. Please refer to Kendall et al. 2018, Sener and Koltun 2018, Lin et al. 2019 and Lin et al. 2020, cited in our introduction section.
>
> **7. “The computation of the HDR are not discussed.”**
>
> The computation of HDR is an established standard procedure, which we explain right after Theorem 4 and cite Juan et al. 2022 for an R package that implements it. This is also not our focus in this paper. In practice, we use Gaussians to model the parameter distributions, and we select an interval, with the lower and upper bounds symmetrically on the two sides of the mean, such that the enclosed probability density function has an integral = 1 - $\alpha$. The two bounds are computed as the inverse cumulative density function (icdf) of a Gaussian at the quantiles of 0.5 - (1 - $\alpha$) / 2 and 0.5 + (1 - $\alpha$) / 2. A Gaussian’s icdf can be very fast computed by tools like scipy.

---

> > ### Author Response · Authors · 2023-11-22
> >
> > Then, on the questions:
> >
> > **8. "What advantages does one distance metric over another for similarity among distributions?"**
> >
> > We have discussed the advantages of choosing the 2-Wasserstein metric in Appendix D. That is, we can obtain this distance efficiently by computing the distance among two distributions’ parameters, when the two distributions are Gaussian.
> >
> > **9. "How do the authors ensure a fair comparison between rehearsal-based methods and IBCL (in terms of buffer growth)?"**
> >
> > We would like to correct the reviewer that rehearsal-based methods, like GEM and A-GEM, do not have fixed-size buffers. Instead, they use **fixed-size buffers per task**, i.e., they cache a number of data points per task, and the total buffer size grows along with the number of tasks. This can be found in Section 3 of the GEM paper (Lopez-Paz and Ranzato 2017, as cited in our paper). Therefore, both rehearsal-based and IBCL grow their buffer along the number of tasks, leading to a fair comparison.
> >
> > **10. “ The L2P is a replay-free model, which is not a fair comparison since the model used the replay buffer.”**
> >
> > Please refer to our discussion with reviewer 79Li. We added this new baseline in our revised manuscript because we have been criticized for not comparing IBCL to “more efficient methods”, such as prompt-based methods. Indeed, there is no established way to specify a preference over task to L2P and other prompt-based methods, and we only made an attempt to do so, as explained in Section 5.1. The buffer growth rate is linear for L2P, since it trains more prompts as we have more number of tasks, but this coefficient of linear growth is very small, because each prompt only takes a very small memory. We do not quite understand “how does the performance change with the sublinear growth”, because L2P does not have a sublinear growth. We will be grateful if the reviewer can clarify this question.
> >
> > Questions 2-5 are already answered in our bullet points 3, 6, 7 and 4, respectively.
> >
> > Again, we appreciate the reviewer for the thoughtful comments, and we hope our answers can lift the confusions. We would be grateful if you can consider raising our rating accordingly based on our response. Thank you in advance.

---

> > > ### Comment · Reviewer_6n6n · 2023-11-27
> > >
> > > Thanks to the authors for the response.
> > >
> > > The responses could be more satisfactory and partially address the concerns.
> > >
> > > The Pareto-optimality motivation for CL needs to be clarified, and other replay approaches show better results with constant buffer size.
> > >
> > > There are many recent works with constant buffer size that need to be discussed, as well, as the baselines need to be stronger.
> > >
> > > Comparisons to recent replay based baselines [1, 2, 3, 4, 5], should be reported.
> > >
> > > Moreover, the finitely generated credal set (FGCS) is not novel and hence not a contribution of this paper.
> > >
> > > Also, how does Pareto-optimality in continual learning differ from multi-task learning in general? In other words, what advantage the Pareto-optimal model has specifically in the CL problem needs to be discussed.
> > >
> > > The replay-based approaches [1-5] have a fixed/constant buffer size, say M. Now, as per this paper, if the replay buffer size increases dynamically with time, it can exceed M. Hence, the result can be worse than the fixed buffer size M. This is not supported experimentally.
> > >
> > > The authors should clarify this point.
> > >
> > > The authors should compare their method with the replay-based methods [1-5] with a fixed buffer size M and show that their method is better than these methods.
> > >
> > > Considering these points, I will keep my rating.
> > >
> > > 1. Lin, Huiwei, Baoquan, Zhang, Shanshan, Feng, Xutao, Li, Yunming, Ye. "PCR: Proxy-based Contrastive Replay for Online Class-Incremental Continual Learning." Proceedings of the IEEE/CVF Conference on Computer Vision and Pattern Recognition (CVPR). 2023.
> > > 2. Fahad Sarfraz, Elahe Arani, Bahram Zonooz. "Error Sensitivity Modulation based Experience Replay: Mitigating Abrupt Representation Drift in Continual Learning." The Eleventh International Conference on Learning Representations (ICLR). 2023.
> > > 3. Jeeveswaran, Kishaan, Prashant, Bhat, Bahram, Zonooz, Elahe, Arani. "BiRT: Bio-inspired Replay in Vision Transformers for Continual Learning." International Conference on Machine Learning (ICML). 2023.
> > > 4. Nie, Xing, Shixiong Xu, Xiyan Liu, Gaofeng Meng, Chunlei Huo, and Shiming Xiang. "Bilateral Memory Consolidation for Continual Learning." In Proceedings of the IEEE/CVF Conference on Computer Vision and Pattern Recognition (CVPR). 2023.
> > > 5. Wang, Zhenyi, Li Shen, Donglin Zhan, Qiuling Suo, Yanjun Zhu, Tiehang Duan, and Mingchen Gao. "MetaMix: Towards Corruption-Robust Continual Learning With Temporally Self-Adaptive Data Transformation." In Proceedings of the IEEE/CVF Conference on Computer Vision and Pattern Recognition (CVPR). 2023.

---

> > > > ### Author Response · Authors · 2023-11-27
> > > >
> > > > Thank you for your response.
> > > >
> > > > 1. More rehearsal-based baselines with fixed buffer size need to be compared.
> > > >
> > > > We thank the reviewer for pointing these papers out. A fixed buffer size is not comparable to our approach, because there is no established methods to specify preferences on these methods. When data from different tasks are all cached, established methods (like Lin et al. 2019) specify a preference by using a weighted sum of losses on these data, where the weight being the preference. However, with fixed buffer size that does not remember which task each data point belongs to, there is no established method to specify this weight. The goal of our experiment that compares to GEM, A-GEM and VCL is to show that IBCL has advantages over **methods with established ways to specify preferences**.
> > > >
> > > > We are willing to make some attempts to specify preferences to the new baselines you recommended, like what we have done to L2P. Although we are afraid the design and experiment cannot be finished before December 1, we are willing to try.
> > > >
> > > > 2. The Pareto-optimality motivation for CL needs to be clarified. How does Pareto-optimality in continual learning differ from multi-task learning in general?
> > > >
> > > > The motivation for Pareto-optimality in CL is the same as for generic multi-task learning. As explained in our section 1 and appendix C, CL is only a special case of multi-task learning, and therefore it has a performance trade-offs among tasks. The motivation for Pareto-optimality is already well-established in the cited papers, such as Lin et al. 2019. Consider there exist multiple tasks, we certainly don't want our learned model is strictly worse than any other model on all tasks, i.e., we want Pareto-optimality. This is the same motivation for optimality in single-task learning.
> > > >
> > > > 3. The finitely generated credal set (FGCS) is not novel and hence not a contribution of this paper.
> > > >
> > > > We never claim that FGCS is a contribution of this paper. Our claimed contribution is using FGCS as a knowledge base representation for CL, and the benefits in performance and efficiency it provides. We also don't think it is fair to reject a paper because it uses some established data structures. Quality papers have been built on "not novel" data structures such as transformers, recurrent networks and convolutional networks.

---

### Author Response · Authors · 2023-11-17
**Please read our revised manuscript**

Dear reviewers, we appreciate your thoughtful comments and would like to invite you to read our revised manuscript, which significantly improves readability and addresses your concerns.

In our latest manuscript, we theoretically and empirically show that IBCL achieves high continual learning performance under specified preferences with low training overhead, improving on both rehearsal-based and prompt-based continual learning algorithms.

Modifications in the revised manuscript include but are not limited to
1. A simplified version of introduction, including a more readable story, a simpler workflow figure of IBCL and the corresponding explanation.
2. Moving more relevant information to the main text, and less relevant one to the appendix.
3. Simpler and more consistent use of notations.
4. Avoiding very complex explanations and jargons for technical details.
5. A simpler explanation of how our experiments work.

Moreover, our experiments compare IBCL to two new baselines suggested by Reviewer 79Li: A-GEM [1] and L2P [2]. The results show that when preferences over tasks are specified, IBCL improves on rehearsal-based methods, like A-GEM, showing better performance as well as much higher efficiency, with a constant training overhead not scaling up with number of preferences. Although prompt-based methods, like L2P, are also efficient in the sense of having  constant overheads, IBCL has much higher continual learning performance.

[1] Chaudhry, Arslan, et al. "Efficient lifelong learning with a-gem." arXiv preprint arXiv:1812.00420 (2018).

[2] Wang, Zifeng, et al. "Learning to prompt for continual learning." Proceedings of the IEEE/CVF Conference on Computer Vision and Pattern Recognition. 2022.

---

### Author Response · Authors · 2023-11-27
**Hope for more discussions with reviewers**

Dear reviewers, as the discussion period of this paper is extended to the end of December 1, we sincerely hope we can all engage in more discussions, especially now that the CVPR supplementary material deadline is past. Thank you in advance.

---

### Author Response · Authors · 2023-11-28
**Clarification on our experiment baseline selection**

Dear reviewers and ACs,

We would like to clarify the reason for our current experiment baseline selection. We understand there are many novel and more efficient continual learning algorithms. However, since our tool, IBCL, is designed for continual learning **under a specified preference over tasks**, we need to compare to **state-of-the-art algorithms that allows specifying this preference**. Specifically, state-of-the-art continual learning algorithms specify a preference by using a weighted sum over loss functions, i.e., at a task $i$,

$$Loss = \sum_{j=1}^i w_jLoss_j$$

Here, $Loss_j$ is the loss on the training data of a previous task $j$, and the regularization weight $w_j$ is the preference of this task. This approach is used in state-of-the-art publications, cited as Lin et al. 2019, Lin et al. 2020, Ma et al. 2020, Gupta et al. 2021 and Mahapatra and Rajan 2020 in our paper (see our Appendix C). To use this state-of-the-art approach, we need to be able compute $Loss_j$ at a later task $i$, so some training data from task $j$ needs to be cached, i.e., this approach only applies to rehearsal-based methods with data from all previous tasks are available.

Therefore, we choose several rehearsal-based methods, GEM and A-GEM, as well as adding a rehearsal cache to VCL (as done by Servia-Rodriguez et al. 2021, cited in our paper), for comparison. These baselines allow us to specify a preference using the state-of-the-art approach described above.

However, after discussion with reviewer 79Li, we believe adding additional comparison to newer and more efficient continual learning baselines would be beneficial. Based on the reviewer's recommendation, we choose L2P, a prompt-based method. Although the state-of-the-art preference specification technique does not apply to L2P, since it is not rehearsal-based, we made an attempt to specify a preference by adding weights to the trainable prompts. This is merely an attempt and does not belong to any state-of-the-art technique.

Our experiment show that, comparing to the comparable methods (continual learning that allows specifying task preferences), our method IBCL achieves not only high accuracy but also low training overhead, measured by number of batch updates needed.

---

### Meta-Review · Area_Chair_FeN8 · 2023-12-05

**Metareview:**

This paper considers a special setting of continual learning where we allow user-specified preferences on tasks' performances. This is an interesting problem in itself and in general the reviewers found it interesting.

The paper had divided opinions with Reviewer bswF and Reviewer SCN6 leaning towards acceptance but Reviewer 79Li and Reviewer 6n6n learning strongly towards rejection expressing concerns about baselines and evaluation.

There were also concerns regarding the preference scenarios not being extensive enough.

The authors provided a detailed response which was considered. However, in the end, consensus could not emerge for accepting the paper. Some of the reviewers who were initially leaning towards acceptance also expressed their reservations regarding evaluation during the discussion.

The paper indeed has good potential but, due to the issues pointed out regarding evaluation, I find it difficult to recommend acceptance for this paper in its current form. I must admit that it was a difficult decision. The authors are encouraged to include the evaluations suggested by the reviewers and resubmit the work to another venue.

**Justification For Why Not Higher Score:**

The reviewers pointed out several issues regarding evaluation.

**Justification For Why Not Lower Score:**

N/A

---

### Decision · Program_Chairs · 2024-01-16

Reject